# The Twisting and Untwisting of Actin and Tropomyosin Filaments Are Involved in the Molecular Mechanisms of Muscle Contraction, and Their Disruption Can Result in Muscle Disorders

**DOI:** 10.3390/ijms26146705

**Published:** 2025-07-12

**Authors:** Yurii S. Borovikov, Maria V. Tishkova, Stanislava V. Avrova, Vladimir V. Sirenko, Olga E. Karpicheva

**Affiliations:** 1Institute of Cytology, Russian Academy of Sciences, 4 Tikhoretsky Av., 194064 St. Petersburg, Russia; mariiatiskova@gmail.com (M.V.T.); avrova@rambler.ru (S.V.A.); sirw@mail.ru (V.V.S.); karpolka@bu.edu (O.E.K.); 2Department of Pharmacology, Physiology and Biophysics, Chobanian and Avedisian School of Medicine, Boston University, 700 Albany St., Boston, MA 02118, USA

**Keywords:** actin–myosin interaction, conformational changes, muscle contraction, mutations in tropomyosin, ATPase activity of myosin, muscle fiber, Ca^2+^-sensitivity, polarized fluorescence microscopy

## Abstract

Polarized fluorescence microscopy of “ghost” muscle fibers, containing fluorescently labeled F-actin, tropomyosin, and myosin, has provided new insights into the molecular mechanisms underlying muscle contraction. At low Ca^2+^, the troponin-induced overtwisting of the actin filament alters the configuration of myosin binding sites, preventing actin–myosin interactions. As Ca^2+^ levels rise, the actin filament undergoes untwisting, while tropomyosin becomes overtwisted, facilitating the binding of myosin to actin. In the weakly bound state, myosin heads greatly increase both the internal twist and the bending stiffness of actin filaments, accompanied by the untwisting of tropomyosin. Following phosphate (Pi) release, myosin induces the untwisting of overtwisted actin filaments, driving thin-filament sliding relative to the thick filament during force generation. Point mutations in tropomyosin significantly alter the ability of actin and tropomyosin filaments to respond to Pi release, with coordinated changes in twist and bending stiffness. These structural effects correlate with changes in actomyosin ATPase activity. Together, these findings support a model in which dynamic filament twisting is involved in the molecular mechanisms of muscle contraction together with the active working stroke in the myosin motor, and suggest that impairment of this ability may cause contractile dysfunction.

## 1. Introduction

Muscle contraction is driven by the cyclic interaction of myosin cross-bridges, actin filaments, and ATP [1,2]. The cross-bridges repeatedly bind to actin filaments [3,4,5,6], passing through several conformations, including the so-called “strong” and “weak” binding states [7,8,9,10].

The strong binding state is associated with the actomyosin complex in the absence of nucleotides (the AM stage, where A—actin and M—myosin), and in the presence of ADP (the AM•ADP stage of the ATP hydrolysis cycle). They are characterized by a high affinity of myosin for actin [11,12], the slow kinetics of myosin head attachment and detachment from actin filaments [13,14], and the ability of regulated actin to undergo cooperative activation. By contrast, weak binding occurs in the presence of ATP (the AM•ATP and AM•ADP•Pi stages of the ATP hydrolysis cycle, where Pi—inorganic phosphate). This is characterized by myosin having a low affinity for actin [8,9]. Myosin heads attach to, and detach from, F-actin with remarkable rapidity [7,8,9]. There is currently no evidence of the cooperative activation of regulated actin filaments [15]. Based on these differences, it has been suggested that the structure of the “weakly” bound actomyosin complex significantly differs from that of the “strongly” bound complex. It is postulated that the force generation in the actomyosin complex occurs as a result of structural changes that take place during the transition from weak to strong binding [13,16].

Significant progress has been made in elucidating the changes in the myosin motor that occur prior to its dissociation from actin in the presence of ATP, as well as the subsequent conformational changes, known as the recovery stroke, which facilitate ATP hydrolysis. After ATP hydrolysis, the myosin motor enters a pre-power stroke state where it is prepared to produce force and move along actin, with MgADP and Pi being trapped inside the motor. The myosin head contains all of the necessary components for force and movement generation [17]. Notably, transitions between weak and strong binding states are accompanied by enhanced intramolecular movements, not only in myosin, but also in actin. Actin-related regulation is carried out by the actin-binding protein complex tropomyosin–troponin (Tpm–Tn), which is sensitive to changes in intracellular Ca^2+^ concentration. At low Ca^2+^ concentrations, the C-terminal domain of the inhibitory subunit of troponin (TnI) attracts tropomyosin, causing it to bend and then pivot towards the TnI [18]. This “blocked” position of tropomyosin results in steric hindrance, preventing the binding of myosin to actin and inhibiting the cross-bridge cycle during contraction. Recent studies have shown that tropomyosin can pivot relative to fixed points on actin, accompanying structural transitions from the blocked (B) to the closed (C) state [19]. This transition is activated by an increase in Ca^2+^ concentration, leading to the displacement of the inhibitory C-terminal domain of TnI from the actin–tropomyosin complex [20]. Changes in the binding of TnI to actin are accompanied by the shift of tropomyosin approximately 15 degrees to a closed position, which eliminates steric hindrance and enables the weak binding of myosin heads to the actin surface, now partially exposed [20,21,22,23,24]. The isomerization of myosin heads into a strong complex with actin causes the further movement of tropomyosin by approximately 25 degrees into an open position. This configuration of tropomyosin enables neighboring myosin heads to bind, fully activating the thin filament [25,26].

Myosin-generated movement and force can be significantly regulated by actin. Actin catalyzes the sequential release of Pi and MgADP from the active site of myosin. In vitro experiments have shown that changes in the flexibility of actin filaments can contribute to force generation [27,28]. Furthermore, the transition from the strong to the weak binding form is accompanied by noticeable changes in the flexibility of actin filaments [29,30]. Additionally, the binding of myosin to F-actin affects the intramolecular movements of certain regions of the actin polypeptide chain, depending on the nature of the actin–myosin interaction. It has been suggested that the high flexibility and internal mobility of actin, induced by myosin binding, allows for actin to coordinate forces from heads in different conformational states [31]. Further studies have shown that the interaction between the myosin head and actin can induce changes in the helical structure of actin [32], the rotational movements of actin monomers in F-actin during the simulation of various stages of muscle contraction [28,29,33,34,35,36,37], as well as alter the orientation of actin monomers in thin filaments, leading to shifts within the actin monomer itself, including the rotation of subdomain 1 relative to subdomain 2 [38,39] and shifts in the actin subunit, particularly the movement of the small domain of actin [40]. The rotation of monomers may occur as a result of the twisting or untwisting of the actin filament, which may determine the ability of actin filaments to interact with myosin. Thus, the weak binding of S1 causes the actin to straighten and stretch, which activates the ability of the actin filament to bind more strongly with myosin [41]. The binding of tropomyosin and troponin can also influence the twisting of F-actin [41,42,43,44].

Changes in the conformation of actin are clearly cooperative in nature [39,45,46,47] and may play a key role in the mechanisms of muscle contraction [30,35,46,47,48,49]. Despite the significant progress that has been made in understanding the structural changes that actin induces for force and movement generation, the precise role of actin in these processes remains poorly understood [10,50,51,52,53].

To clarify the unknown aspects of muscle contraction mechanisms, studying the effects of mutations on contractile function represents a valuable approach. Normal muscle-contraction mechanisms are disrupted by mutations in sarcomeric proteins, including point mutations or deletions in α-, β-, and γ-tropomyosins (encoded by the genes *TPM1*, *TPM2*, and *TPM3*, respectively [54]). Mutations in these genes lead to a wide range of neuromuscular and cardiac diseases with clinically, histologically, and genetically heterogeneity, including nemaline myopathy, cap myopathy, distal arthrogryposis, congenital fiber type disproportion (CFTD), and others. These diseases are often accompanied by muscle weakness and hypotonia [55,56,57,58,59,60].

A number of tropomyosin point substitutions associated with congenital myopathies (Gln147Pro, Glu117Lys, and Arg91Gly in β-tropomyosin, and Glu150Ala, Arg167His, and Glu173Ala and in γ-tropomyosin, and others) affect amino acid residues that are conserved throughout the length of the tropomyosin molecule. These charged residues electrostatically interact with specific actin residues [57,58]. Molecular dynamics simulations have shown that the reduction in charge due to these substitutions and deletions can lead to the distortion of the energy landscape of the tropomyosin–actin interactions [61], increasing the distance between tropomyosin and the actin filament, as well as altering the position of tropomyosin on actin when its contact with actin is disrupted [60].

In this study, polarization microfluorimetry was used to assess conformational changes in muscle fibers reconstructed with tropomyosin, troponin, and S1 at various stages of the ATPase cycle, with fluorescent labels on actin, tropomyosin, and myosin. Changes in the twisting and bending stiffness of actin and tropomyosin filaments were detected, as well as the tilt of the myosin head relative to the axis of the actin filaments, which correlated with ATPase activity levels. We show that point mutations, such as Arg167His, Lys168Glu in Tpm1.1, Arg91Gly, Glu117Lys, and Gln147Pro in Tpm2.1 and Tpm2.2, Arg90Pro, Glu150Ala, and Glu173Ala in Tpm3.12, significantly alter the twisting of tropomyosin and actin, as well as their bending flexibility. This, in turn, leads to an increase in the calcium sensitivity of thin filaments and a decrease in actomyosin ATPase activity. These results suggest that the twisting and untwisting of tropomyosin and actin filaments, as well as changes in their bending stiffness, play a key role in the molecular mechanisms of muscle contraction.

## 2. Results and Discussion

### 2.1. Ghost Muscle Fibers Reconstituted with Labeled Proteins as a Model for Studying Conformational Changes During the ATPase Cycle

A model system of thin filaments with bound myosin heads was reconstructed in actin-containing ghost fibers. Endogenous proteins, including tropomyosin, troponin, and myosin, were extracted from muscle fibers prior to reconstruction, and the resulting ghost fibers consisted of 80–90% F-actin (see Materials and Methods). The thin filaments were reconstituted with exogenous wild-type (WT) or mutant tropomyosin (Tpm), troponin, followed by decoration with myosin subfragment-1 (S1), under conditions mimicking different stages of the ATP hydrolysis cycle. This approach was previously described [30,48,61,62,63]. Control experiments, including SDS–PAGE analysis and fluorescence intensity measurements of the probes, demonstrated that the relative amount of each incorporated protein was virtually identical across all experiments.

To investigate how the binding of the tropomyosin–troponin complex to actin, the presence of Ca^2+^ ions, and point substitutions in α-, β-, or γ-tropomyosin affect the conformational state of F-actin, tropomyosin, and myosin heads, polarized fluorimetry was employed [48,64,65]. The polarized fluorescence of the analyzed proteins reflects the average structural state of the population of protein molecules [30,48,62,63]. The AM stage of the actomyosin complex was simulated in the absence of nucleotides. Mg–adenosine diphosphate (MgADP) and Mg–adenosine triphosphate (MgATP) were used to mimic the AM•ADP, and AM*•ATP stages, respectively [37,63,66].

F-actin, tropomyosin, and S1 were labeled with fluorescent probes. Fluorescein isothiocyanate conjugated with phalloidin (FITC-phalloidin) was bound to F-actin in the actin groove (FITC-actin); 5-iodoacetamidofluorescein (5-IAF) was covalently linked to Cys residues of Tpm1.1, Tpm2.1, Tpm2.2, and Tpm3.12 (AF–Tpm); N-(iodoacetaminoethyl)-1-naphthyl-amine-5-sulfonic acid (1,5-IAEDANS) was specifically linked to Cys707 of S1 (AEDANS–S1). Incorporation of these proteins initiated polarized fluorescence, the measurement of which allowed for tracking changes in the spatial arrangement and flexibility of F-actin, tropomyosin filaments, and myosin heads during different stages of the ATPase cycle (Figure 1) [30,37,62,67,68].

Modification of Cys707 using a fluorescent probe may affect certain aspects of myosin behavior, but remains an effective method for obtaining information on actin–myosin interactions. It has been shown that labeling of Cys707 reduces the ATPase activity of myosin and the sliding velocity of actin filaments over myosin [69,70]. This modification also diminishes the rotation of the converter domain of the myosin head that takes place during the ATPase cycle [71]. However, experiments conducted in the laboratory of Prof. F. Morales demonstrated that modification of Cys707 with 1,5–AEDANS does not significantly affect the conformational changes in myosin or its ability to generate force. It is important to note that this represented the first demonstration of the cyclic work of cross-bridges and the correlation between conformational changes in myosin heads and force development during muscle contraction, which was achieved using fluorescent probes. The impact of labeling on myosin functional properties depends on various factors, including the probe type, modification method, dye used, and manufacturing conditions under which the dye was synthesized. The 1,5–IAEDANS probe was applied using the method developed by Borejdo and Putnam, and it was shown not to significantly affect the functional properties of myosin heads [72]. In our control experiments, it was also found that the modification did not significantly affect the binding of S1 to actin, either in the absence of nucleotides or in the presence of MgADP (in strong actomyosin binding) or MgATP (in weak binding). Thus, within the experimental design employed in this study, AEDANS–S1 can be utilized to investigate actin–myosin interactions during the ATPase cycle. Changes in binding were evaluated based on changes in the orientation and mobility of the myosin head [67].

It has been established that phalloidin enhances the rigidity of the actin filament [73]. No effect of phalloidin on the actin-activated ATPase activity of skeletal muscle myosin was observed on isolated actomyosin preparation [74] with a 25% increase in ATPase activity observed at pCa 8 on myofibrils, and no increase at pCa 4 [75]. Our control experiments demonstrated that FITC-phalloidin had no effect on the ATPase activity of S1 [48,76]. In light of these findings, we postulate that the effect of FITC-phalloidin on the contractile apparatus of striated muscle is negligible. Modification of tropomyosin by 5–IAF appears to have no substantial impact on the functional characteristics of this protein [77,78], not hindering its binding to actin and the activation or deactivation of thin filaments depending on the calcium concentration in the muscle fiber. Thus, the observed alterations in the conformational state of the labeled actin, tropomyosin, and myosin heads are likely to reflect those occurring during muscle contraction. It is important to note that, in our steady-state experiments, the polarized fluorescence of the studied protein reflects the average structural state of the entire protein population.

It is known that the structure of tropomyosin is pre-shaped for binding with actin [79]. At the same time, during the contraction cycle, actin can change its conformation, rigidity, and twist angle [31,32,80]. Tropomyosin also undergoes conformational changes, moving along the actin surface. We hypothesize that changes in the bending stiffness of both proteins occur in a coordinated manner and are associated with the twisting of the filaments. To assess the contribution of these changes to the ATPase cycle and force generation, we measured the orientation of fluorescent probes on tropomyosin and actin, as well as their bending stiffness in the presence and absence of troponin, calcium ions, and nucleotides. The incorporation of AF-Tpm of α-, β- or γ-isoforms or FITC-actin into ghost fibers, as well as the binding of AEDANS–S1 to F-actin initiated polarized fluorescence [30,37,48,63,67,76]. The polarized fluorescence intensity of four components was measured when the fiber was aligned parallel (IIIII, III⊥) and perpendicular (⊥I⊥, ⊥III) to the excitation light’s polarization plane (see Materials and Methods). Employing the helical plus isotropic model [48] and the fluorescence polarization data allowed us to determine the angle between the emission dipoles and the fiber axis (Φ_E_), the bending rigidity (ε) for AF-Tpm and FITC-actin, as well as the relative number of disordered probes (N), along with Φ_E_ for AEDANS–S1.

### 2.2. Ca^2+^-Mediated Structural Remodeling of Thin Filaments

It is known that F-actin and tropomyosin interact through electrostatic forces [81], enabling independent assessments of their rigidity [30,81,82,83]. Data obtained using probes on α-, β-, and γ-tropomyosins and probes on actin show that the bending stiffness value of tropomyosin in the F-actin–Tpm–Tn complex at high Ca^2+^ levels is significantly higher than that of bare F-actin (Figure 2b,d).

The dependence of fluorescence parameters for FITC-actin, AF-Tpm, and AEDANS–S1 on Ca^2+^ levels has previously been observed, regardless of the tropomyosin isoform (α, β, or γ) incorporated into thin filaments [62,84,85,86,87]. Since FITC-phalloidin binds strongly and specifically to F-actin [38], the changes in Φ_E_ and ε values reflect, respectively, the spatial rearrangements of actin monomers within the actin filament and alterations in their bending stiffness [30,37,48]. The location of the FITC label allows for the observation of conformational changes at the level of the actin double helix. The F-actin filament is organized as a right-handed double helix made up of two single left-handed filaments. It has been shown that the increase in intrinsic twisting or overtwisting of F-actin occurs in a clockwise direction, while untwisting happens counterclockwise [88,89]. Similarly, as AF specifically binds to the Cys residues of tropomyosin, the changes in Φ_E_ and ε values reflect both the spatial rearrangement and the changes in bending stiffness of tropomyosin strands [37].

According to Figure 2, the binding of troponin to FITC-actin in the presence of wild-type tropomyosin at high Ca^2+^ levels leads to an increase in the Φ_E_ value and a decrease in the ε value (ε_Actin_). The extent of these changes varied depending on the tropomyosin isoform (α, β, or γ) (Figure 2a,b), likely reflecting differences in their amino acid composition, and may be attributed to the global conformational transitions of actin monomers within the thin filaments [30,37,63,68,86]. These conformational rearrangements occur cooperatively along the filament [66].

Since FITC-phalloidin binds within the groove formed by three adjacent actin subunits [90,91], alterations in the Φ_E_ value may reflect changes in the helical structure of the actin filament. This may include variations in the pitch of the short- and long-pitch helices [32,92]. An increase in the Φ_E_ value at high Ca^2+^ levels has also been reported for fluorescent probes localized in various regions of subdomain 1 of the actin monomer, including those bound to Cys374, Cys343, Cys10, Lys373, Lys61, and Glu41 [48]. The observed changes in the Φ_E_ value for FITC-actin can be explained by the rotation of entire actin subunits or their major domains outward from the filament axis [30,35,37,62]. This rotation is accompanied by a decrease in F-actin bending stiffness (Figure 2b), indicating the untwisting of actin filaments—previously observed at high Ca^2+^ levels [73].

A significant increase in Φ_E_ values and a decrease in ε values are observed during simulations of thin filament activation, indicating conformational changes that shift F-actin monomers into the so-called “ON” state (switched-on) [30,93]. In this state, actin monomers activate myosin ATPase [30,37,76,87], whereas in the “OFF” (switched-off) state, this process is inhibited. The two states of actin monomers are in rapid equilibrium, and the proportion of monomers in each state can be modulated by the binding of tropomyosin, troponin, and myosin heads to F-actin, as well as by certain small molecules (Figure 2a,b) [67].

Ca^2+^ binding to the FITC–actin–Tpm–Tn complex leads to an increase in Φ_E_ values (Figure 2a). This indicates that Ca^2+^ promotes the transition of a larger population of actin monomers into the ON state [62,67]. Since FITC-phalloidin binds specifically to amino acid residues located within or near the myosin binding site on actin, it has been proposed that alterations in the configuration of this site on F-actin, initiated by regulatory proteins or by myosin head binding, could be reflected by changes in the orientation of the FITC probe’s oscillators dipoles [30,66,93,94]. Accordingly, differences in the structural and functional states of actin monomers involve differences in the configuration of the myosin binding site. In the ON state, actin monomers readily support strong myosin binding, while in the OFF state, such binding is inhibited.

The binding of troponin to the F-actin–AF–Tpm complex at high Ca^2+^ levels leads to a decrease in Φ_E_ value and an increase in ε_Tpm_ values (Figure 2c,d, Table 1). As proposed earlier, these change in AF–Tpm occurs alongside global conformational alterations in the tropomyosin strands [30,37,68,86]. This is accompanied by the rotation of the fluorescent probe attached to tropomyosin towards the thin filament axis and by an increase in the bending stiffness of the tropomyosin strands, potentially as a response to the concurrent reduction in F-actin stiffness [30,62,87].

The rigidity of both actin [92] and tropomyosin [95,96] filaments appears to helical conformation. Untwisting of either filament is associated with decreases bending stiffness, whereas increased helical twisting enhances filament stiffness [95,96] (Figure 2, Table 1). Troponin appears to modulate filament stiffness and helical conformation in a Ca^2+^-dependent manner.

Thus, when modeling the C-state of thin filaments, binding of the troponin–tropomyosin complex induces conformational changes in actin that result in the rotation of the fluorescent probes, untwisting of actin filaments, and twisting of tropomyosin filaments. Actin untwisting may be associated with TnI dissociation, whereas tropomyosin twisting (or overtwisting) likely occurs upon TnI binding to actin [95,96]. Given the tight and specific binding of the fluorescent probes, it can be assumed that actin and tropomyosin filaments rotate in opposite directions during the transition to the C-state, leading to their relative sliding and forming a configuration that facilitates strong binding of myosin heads to actin. It should be noted that fibers containing different tropomyosin isoforms respond to Ca^2+^ activation with varying sensitivity. For instance, previous studies have shown higher Ca^2+^ sensitivity in complexes containing β-tropomyosin compared to α-tropomyosin [97]; indeed, changes in fluorescence parameters observed after the addition of high Ca^2+^ to the FITC–actin–Tpm–Tn complex were most pronounced in the β-tropomyosin-containing complexes.

The transition to the C-state is accompanied by a decrease in the bending stiffness of actin filaments ε_Actin_ and an increase in the bending stiffness of tropomyosin strands ε_Tpm_ (Table 1). For all tropomyosin isoforms studied, the bending stiffness was found to be from two to three times higher than that of actin filaments. The greatest difference in the ε_Tpm_/ε_Actin_ ratio was observed for γ-tropomyosin (see Figure 2b,d and Table 1). The observed decrease in actin bending stiffness correlates with an increased proportion of actin monomers in the ON state [30]. Moreover, the increases stiffness of tropomyosin suggests an activation of tropomyosin strands. We hypothesize that, in the C-state, both actin and tropomyosin are conformationally activated and arranged in a manner that promotes strong binding of myosin heads to F-actin.

Conversely, when modeling the B-state in thin filaments, a decrease in Ca^2+^ concentration from 10^−4^ M (pCa 4) to 10^−8^ M (pCa 8) and the dissociation of Ca^2+^ from troponin C (TnC) induces the attachment of TnI to actin monomers. This interaction leads to the twisting of actin filaments and the untwisting of tropomyosin strands, which are accompanied by the decreased bending stiffness of tropomyosin and increased stiffness of actin filaments (Figure 2b,d). These changes are associated with reduced Ca^2+^ affinity for troponin [30,83]. For all tropomyosin isoforms in the B-state, bending stiffness was significantly lower than at high Ca^2+^, whereas the stiffness of actin filaments was higher at low Ca^2+^ (Figure 2b, Table 1). This suggests that actin and tropomyosin adopt a relative arrangement that decreases the number of switched-on actin monomers. Tropomyosin is known to pivot around fixed points on actin to facilitate transitions between the B- and C-states. At low Ca^2+^ levels, TnI domains appear to attract tropomyosin, causing them to bend and pivot, thereby blocking myosin binding and preventing the cross-bridge cycle [19]. Our data suggest that opposing twisting forces from actin and tropomyosin cause tropomyosin to slide along the actin’s surface relative to the myosin binding site (Figure 3). This displacement limits myosin binding through both allosteric (via conformational changes in actin) and steric (via unfavorable spatial organization) means. For example, the twisting of actin filaments may directly alter the configuration of the myosin binding site.

Since actin and tropomyosin interact via electrostatic forces [97], it can be hypothesized that their relative twisting and untwisting, as well as changes in their bending stiffness, may affect the spatial organization of their interaction surfaces. This, in turn, could alter the nature and the number of electrostatic bonds between them. This interpretation is consistent with the data on changes in the energy landscape of actin–tropomyosin interactions: during the transition from the B-state to the myosin-bound (M) state, the number of actin residues available for tropomyosin binding decreases more than twofold [61].

These mechanisms, regulated by troponin and Ca^2+^ levels, may play an important role in regulating the interaction between actin and tropomyosin and may also affect the structure and function of the myosin binding site on actin. At high Ca^2+^ levels, such regulation may activate actin monomers, whereas at low Ca^2+^ levels, it may lead to their deactivation. Additionally, the increased flexibility of tropomyosin at low Ca^2+^ levels may promote the formation of additional electrostatic bonds with actin, potentially enhancing the oscillatory movement of tropomyosin over the myosin binding site. These oscillations may sterically obstruct myosin head attachment to F-actin.

We hypothesize that the molecular mechanisms regulating muscle contraction are based on the coordinated twisting and untwisting of actin and tropomyosin filaments. At low Ca^2+^ levels, TnI binds to actin filaments, inducing their twisting and increasing their bending stiffness. These configurational changes in the myosin binding site hinder the attachment of myosin heads. The enhanced flexibility of tropomyosin under these conditions may allow for it to adopt fluctuating positions in front of the myosin binding site, sterically interfering with myosin access. At high Ca^2+^ levels, actin filaments untwist, and their bending stiffness decreases. At the same time, tropomyosin filaments twist and become more rigid. The structural changes in the configuration of the myosin binding site and the untwisting of actin filaments enable to transition into the C-state, facilitating the weak binding of myosin heads.

The subsequent attachment of myosin heads to thin filaments at elevated Ca^2+^ concentrations further reinforces actin untwisting and tropomyosin twisting, accompanied by changes in their bending stiffness. These conformational shifts lead to the sliding of tropomyosin strands relative to actin. The direction of this displacement may depend on Ca^2+^ concentrations, myosin, and nucleotides (as discussed below).

### 2.3. Regulation of Actin–Tropomyosin Twisting and Myosin Head Orientation by S1, Nucleotides, and Ca^2^

The binding of S1 to the F-actin–Tpm–Tn complex significantly affected the polarized fluorescence parameters of FITC–Actin and AF–Tpm, depending on the Ca^2+^ levels and regardless of nucleotides presence. AEDANS–S1 also underwent conformational changes upon Ca^2+^ and nucleotide binding. In control experiments lacking troponin and S1, Ca^2+^ and nucleotides had no noticeable impact on FITC-actin or AF-Tpm fluorescence. These findings suggest that the observed changes primarily reflect S1- and Ca^2+^-dependent conformational rearrangements in F-actin and tropomyosin under various ATPase cycle stages [37].

According to Figure 2, S1 binding to F-actin (AM stage), led to decreased Φ_E_ value for AF–Tpm; the ε value decreased for the β-tropomyosin and increased for the α- and γ-tropomyosins at high Ca^2+^ levels. For FITC-actin, the Φ_E_ value increased; the ε value increased in the presence of the α- and β-tropomyosins and decreased in the presence of γ-tropomyosin. These changes in Φ_E_ values suggest that the emission dipoles of FITC-phalloidin on actin and AF on tropomyosin move in opposite directions—toward and away from the center of the thin filament, respectively.

Since FITC-phalloidin and AF bind specifically to their target proteins, and the fluorescence spectra of these probes remain unchanged, the observed changes in the fluorescence parameters of FITC-actin and AF--Tpm likely indicate a positional shift in tropomyosin relative to F-actin monomers. At high Ca^2+^ levels, changes in the AF-Tpm parameters may result from the spatial rearrangement of the tropomyosin helix. An increase in the Φ_E_ value for FITC-actin and its decrease for AF-Tpm at high Ca^2+^ levels suggest that, during simulations of the rigor state (M–state) in muscle fibers, S1 binding to the F-actin–Tpm–Tn complex causes an untwisting of the actin filaments and a concurrent twisting of the tropomyosin strands (Figure 2a,c). This is accompanied by a decrease in the bending stiffness of actin filaments and an increase in the bending stiffness of β- and γ-tropomyosin strands, while α-tropomyosin shows only a slight reduction in the stiffness (Figure 2b,d). In the M-state, the bending stiffness of all tropomyosin isoforms was significantly higher than that of actin filaments in the presence of the corresponding tropomyosins. A decrease in actin filament stiffness correlates with an increase in the number of switched-on actin monomers, while an increase in tropomyosin stiffness may indicate the cooperative activation of tropomyosin molecules along the filament. Thus, in the M-state, the spatial configuration of actin and tropomyosin likely favors the strong binding of myosin heads to F-actin by the optimal rearrangement of the myosin-binding site. As noted above, in the absence of troponin and S1, the addition of Ca^2+^ or nucleotides did not significantly alter the polarized fluorescence parameters. However, the addition of troponin and high Ca^2+^ levels caused the untwisting of actin filaments and the twisting of tropomyosin filaments during the transition from the B-state to the C-state. The subsequent addition of S1 to this complex and the further transition to the M-state led to even more pronounced changes in bending stiffness and twisting characteristics. These findings suggest that thin filament activation by myosin heads is more strongly activation-mediated solely by the Tpm–Tn complex. These findings are supported by data showing that nucleotide-free and ADP-bound myosin promotes tropomyosin displacement along the actin’s surface [96], a rearrangement likely required for the enhanced activation observed in the M-state.

During the transition from the AM*•ATP to the AM stage at high Ca^2+^ levels, the Φ_E_ value increased for FITC-actin, but decreased for both AF–Tpm–WT (Figure 2a,c) and AEDANS–S1 (Figure 4a). In line with our findings [37], the increase in Φ_E_ values for FITC-actin and the decrease for AEDANS–S1 can be interpreted as indicative of an increased relative amount of the switched-on actin monomers and strongly bound myosin heads to F-actin [67]. Furthermore, during this transition, the bending stiffness of FITC-actin decreased, while it increased for the β- and γ-isoforms of AF-Tpm (Figure 2). These changes in fluorescence parameters are associated with the enhanced cooperativity of actin monomer activation. This suggests that, in the AM*•ATP stage at high Ca^2+^, most actin monomers and tropomyosin molecules are in an inactive state, and myosin heads are only weakly bound to F-actin (Figure 4). In contrast, the transition to the rigor (AM) stage promotes the activation of actin monomers and an increase in the amount of strongly bound myosin heads. The observed changes in Φ_E_ and ε values indicate that actin filaments untwist, while tropomyosin filaments twist during the transition to the rigor state. This interpretation is supported by the corresponding changes in the number of randomly oriented fluorophores, N, for AEDANS-S1. In the presence of MgATP, the N values were the highest and decreased in the AM stage (Figure 4b), which may reflect a higher number of S1 strongly bound to F-actin in the presence of MgATP [67].

ATP hydrolysis, a key step in force generation, occurs during the transition from the AM*•ATP to the AM stage. To better understand its role, we analyzed the transition from AM*•ATP to AM•ADP separately. In the presence of MgADP at high Ca^2+^ levels, which mimics the AM•ADP stage of the ATPase cycle, the Φ_E_ values for both FITC-actin and AEDANS–S1 were lower than in the absence of nucleotides (AM stage) (Figure 2a and Figure 4a). Additionally, these measurements reveal that the bending stiffness of actin filaments was higher in the presence of MgADP (Figure 2b). This indicates that the number of activated actin subunits and strongly bound myosin heads in the MgADP-bound complex exceeds that in the MgATP-bound state, but remains lower than in the rigor (AM) state (Figure 4). Overall, the structural changes during the transition from AM*•ATP to AM•ADP are greater than those observed during the subsequent AM•ADP to AM transition, indicating the significant role of ATP hydrolysis in force generation, while ADP release makes a smaller contribution, which is consistent with the difference in the amplitude of myosin lever arm motion [53].

During the transition from the AM*•ATP stage at high Ca^2+^ to the ATP-bound stage at low Ca^2+^ levels (which mimics muscle relaxation, “Relax”), the Φ_E_ value decreased for FITC-actin but increased for AF–Tpm–WT (Figure 2a,c) and AEDANS–S1 (Figure 4a). These changes suggest that actin filaments undergo twisting, while tropomyosin strands untwist, leading to a change in the direction of filament sliding, opposite to what is observed during force generation (Figure 2a,c). Correspondingly, the bending stiffness of these filaments changes: it increases for FITC-actin in the presence of α- and β-tropomyosin, but decreases for AF–α–Tpm and AF–γ–Tpm. We hypothesize that these coordinated changes in Φ_E_ and ε can be explained by the specialized roles of tropomyosin isoforms in muscle fiber function. The α-isoform predominates in fast-twitch fibers, whereas the γ-isoform is characteristic of slow-twitch fibers. The relatively small stiffness changes in AF–α–Tpm may be linked to the rapid contraction–relaxation cycles in fast-twitch fibers, where time constraints limit the extent of structural rearrangement. In contrast, the larger changes observed with γ-tropomyosin may support sustained contractions in slow-twitch fibers (Figure 2b,d). However, the overall pattern of filament twisting and untwisting during transitions between functional states remains consistent across isoforms, likely reflecting their essential role.

Changes in the bending stiffness (ε) of F-actin and Tpm-WT observed at various intermediate stages of the ATPase cycle can be explained by substantial alterations in the amount and nature of electrostatic interactions between F-actin and tropomyosin within the F-actin–Tpm–Tn–S1–nucleotide complex. For instance, the high ε value for tropomyosin observed in the absence of nucleotides or in the presence of MgADP could be attributed to an increase in the number or character of electrostatic interactions between F-actin, S1, and tropomyosin, as previously suggested [80,97]. Given that actin filaments exhibit low bending stiffness in the strongly bound actomyosin intermediate state of the ATPase cycle, while tropomyosin strands showed high stiffness, it seems unlikely that the increased stiffness of tropomyosin is associated with a greater amount of actin–tropomyosin bonds. Another possible explanation is that, at high Ca^2+^ levels, TnI binds to tropomyosin [18]. This interaction may induce conformational changes in tropomyosin, leading to an increased bending stiffness (Figure 2 and Table 2). In addition, the binding of the myosin head to tropomyosin strands may also contribute to the increased bending stiffness of tropomyosin [98,99,100]. Our data are consistent with this assumption (Table 2). Structural changes in tropomyosin may be transmitted to F-actin, causing changes in its conformation as well. Consequently, myosin head binding to both tropomyosin and actin filaments may promote an increase in the number of switched-on actin monomers. As actin and tropomyosin filaments can undergo Ca^2+^- and nucleotide-dependent twisting and untwisting in response to troponin and myosin binding, their bending stiffness likely depends on the nature of these structural changes. Such dynamic reorganizations within the actin–Tpm–Tn complex may play a crucial role in regulating muscle contraction by modulating the ability of actin and tropomyosin for myosin heads, ultimately affecting contractile capacity.

Therefore, at high Ca^2+^ levels, conformational changes in both tropomyosin and actin filaments, as well as in the S1 occur, either with or without nucleotides. As a result, actin filaments exhibit decreased bending stiffness, while tropomyosin filaments become stiffer, indicating their respective untwisting and twisting. This structural configuration facilitates the strong binding of myosin heads to actin filaments and weak binding to tropomyosin strands is possible. In contrast, at low Ca^2+^ levels, the troponin complex, regardless of the presence of S1 and nucleotides, increases the bending stiffness of actin and decreases that of tropomyosin, suggesting that actin becomes more twisted, while tropomyosin untwists. This spatial reorganization of the myofilament hampers the strong binding of myosin heads to both actin and tropomyosin filaments (Figure 2 and Figure 4). Therefore, the regulation of actomyosin interaction during the ATPase cycle involves the coordinated twisting and untwisting of actin and tropomyosin filaments, accompanied by specific changes in their bending stiffness. These processes are further modulated by the structural and functional state of the myosin head.

### 2.4. Twisting and Untwisting of Actin and Tropomyosin Filaments May Be Involved in the Molecular Mechanisms of Muscle Force Production

During the simulations of different stages of the ATPase cycle at low or high Ca^2+^ levels, the bending stiffness of actin and tropomyosin filaments changes significantly. The changes appear to be determined by the character of filament twisting: the untwisting of actin or tropomyosin filaments reduces their bending stiffness, whereas increased twisting enhances it. Nucleotides and Ca^2+^ can modulate this relationship (Figure 2, Figure 5, and Table 2).

During the simulations of different stages of the ATPase cycle, we observed changes in the bending stiffness of actin and tropomyosin filaments (Table 2), along with variations in the extent and direction of their twisting (Figure 2). Changes in myosin head orientation relative to actin filaments (Φ_E_) and changes in their mobility (N) were also observed (Figure 4).

These findings highlight that the components of the contractile system act in a coordinated manner, with myosin heads contributing to structural rearrangements within thin filaments. Specifically, during simulation of the AM*•ATP stage (in the presence of MgATP and Ca^2+^), myosin heads, despite weak electrostatic binding to the actin filament [101,102], decrease the angle Φ_E_ of FITC-actin containing α-, β-, or γ-tropomyosins by 0.7 ± 0.2, 2.0 ± 0.2, and 0.4 ± 0.1 degrees, respectively (*p* < 0.05; Figure 2), which can be interpreted as an increase in actin filament twisting [30]. These results suggest that even the weak binding of myosin heads to actin filaments causes overtwisting, which is accompanied by an increase in the bending stiffness of actin filaments.

The values of ε_Actin_ increase during the transition from the C-state to the AM*•ATP stage (Table 2). In the presence of α-, β-, and γ-tropomyosin, bending stiffness rises from 3.5 × 10^–26^ to 7.6 × 10^–26^ Nm^2^ (by 114%), from 5.1 × 10^–26^ to 7.8 × 10^–26^ Nm^2^ (by 53%), and from 5.2 × 10^–26^ to 5.5 × 10^–26^ Nm^2^ (by 6%), respectively. These data suggest that the weak binding of myosin heads induces the additional twisting of actin filaments, resulting in a marked increase in their bending stiffness.

It is possible that, under the influence of weak electrostatic interactions between myosin heads and actin filaments in the AM*•ATP stage [101,102], tropomyosin undergoes changes in helical twisting. As shown in Figure 2, the Φ_E_ values increase 0.8 ± 0.1 degrees and 1.4 ± 0.2 degrees for AF–α–Tpm and AF–β–Tpm, respectively, and decrease by 0.5 ± 0.2 degrees for AF–γ–Tpm compared to the C-state (*p* < 0.05; Figure 2). These results indicate that α- and β-tropomyosin helices markedly untwist during the transformation from the C-state to the AM*•ATP stage, whereas γ-tropomyosin exhibits minimal change.

The bending stiffness of AF-Tpm, ε_Tpm_ for the α-isoform significantly increases from 9.4 × 10^–26^ to 12.4 × 10^–26^ Nm^2^ (by 44%), while for the β- and γ-isoforms, it decreases from 11.4 × 10^–26^ to 8.7 × 10^–26^ Nm^2^ (by 31%) and from 13.9 × 10^–26^ to 9.9 × 10^–26^ Nm^2^ (by 39%), respectively (Table 2). It can be assumed that, during the AM*•ATP stage, the weak binding of myosin heads to actin filaments causes a reduction in the twisting of tropomyosin filaments, which is accompanied by an increase in their bending stiffness for the α-isoform and by a decrease in the stiffness for the β- and γ-isoforms.

Thus, during the transition from the C-state to the AM*•ATP stage of the ATPase cycle, actin filaments undergo increased twisting, whereas tropomyosin filaments become untwisted. The bending stiffness of actin filaments ε_Actin_ increases, while the bending stiffness of tropomyosin filaments ε_Tpm_ decreases (Figure 2 and Table 2). As a result, the ratio of ε_Tpm_/ε_Actin_ decreases. For the α-, β-, and γ-isoforms, the ratio of ε_Tpm_/ε_Actin_ decreases from 2.65 to 1.64 (by 38%), from 2.23 to 1.10 (by 49.3%), and from 2.67 to 1.80 (by 32%), respectively (Table 2).

Since actin and tropomyosin filaments differ in their direction and degree of rotation, as well as in their bending stiffness, tropomyosin strands likely slide relative to actin filaments during the transition from the C-state to the AM•ATP stage. During this relative sliding, the overtwisted actin filaments transform into a conformational state that is favorable for the weak binding of myosin heads. In this configuration, myosin heads appear to turn away from the actin filament axis (the values Φ_E_ of AEDANS–S1 increase), exhibiting increased mobility (the values of N increase) (Figure 4). Recent time-resolved cryo-electron microscopy (Cryo-EM) data have revealed changes in the orientation and mobility of myosin heads during weak and strong binding states [103]. These findings are consistent with our data and support the model of coordinated actin–tropomyosin remodeling during the early stages of myosin engagement.

In the presence of MgADP and high Ca^2+^, when mimicking the transition from the AM*•ATP to the AM•ADP stage of the ATPase cycle (i.e., during Pi release), myosin heads can simultaneously interact with both actin and tropomyosin filaments [53,96,99,101,103,104]. Our data (Table 2) support this interaction: the binding of myosin in this state induces an increase in the Φ_E_ angle of FITC-actin, indicating the untwisting of the actin filament. Specifically, the Φ_E_ angle increases by 2.7 ± 0.2, 0.9 ± 0.1, and 0.6 ± 0.2 degrees for actin filaments containing α-, β-, and γ-tropomyosin, respectively (*p* < 0.05). Conversely, the Φ_E_ angle for the corresponding tropomyosins strands decreases by 3.5 ± 0.2, 1.4 ± 0.2, and 1.1 ± 0.2 degrees, respectively (Figure 2), indicating filament twisting. These structural changes are accompanied by a sharp decrease in the bending stiffness of actin filaments, in contrast to the increase observed at the AM*•ATP stage (Table 2). In parallel, the bending stiffness of tropomyosin filaments markedly increases for all isoforms, consistent with enhanced tropomyosin twisting at the AM•ADP stage. The potential for twisting and untwisting of tropomyosin filaments on actin filaments has been shown previously [95]. Moreover, our observations of decreased actin stiffness align with prior proposals that actin flexibility is required for increased sliding velocity and for effective coordination of myosin head activity during the ATPase cycle.

During the transition from the AM*•ATP to the AM•ADP stage, the bending stiffness of actin filaments (ε_Actin_), containing α-, β-, or γ-tropomyosin significantly decreases, respectively, from 7.6 × 10^–26^ to 4.2 × 10^–26^ Nm^2^ (by 45%), from 7.8 × 10^–26^ to 7.0 × 10^–26^ Nm^2^ (by 11%), and from 5.5 × 10^–26^ to 5.0 × 10^–26^ Nm^2^ (by 10%). In contrast, the bending stiffness of AF-Tpm strands ε_Tpm_ increases from 12.4 × 10^–26^ to 13.5 × 10^–26^ Nm^2^ for α-Tpm (by 9%), from 8.7 × 10^–26^ to 13.0 × 10^–26^ Nm^2^ (by 49%) for β-Tpm, and from 9.9 × 10^–26^ to 14.3 × 10^–26^ Nm^2^ for γ-Tpm (by 44%) (*p* < 0.05). As a result of these opposing changes, the ratio of ε_Tpm_/ε_Actin_ increases during this transition, rising from 1.6 to 3.2 for α-Tpm (by 98%), from 1.1 to 1.9 for β-Tpm (by 68%), and from 1.8 to 2.8 for α-Tpm (by 57%) (Table 2).

From these data, we can conclude that the transition from the AM*•ATP to the AM•ADP stage is accompanied by coordinated changes in the twisting of actin and the untwisting of tropomyosin filaments, along with corresponding changes in their bending stiffness. These mechanical transformations seem to be accompanied by significant alterations in specific interaction sites between actin and tropomyosin, likely occurring as tropomyosin strands slide relative to the actin filaments during Pi release. Consequently, the ε_Tpm_/ε_Actin_ ratio may serve as an indicator of the performance of the contractile system in muscle fiber (see below).

Due to the distinct mechanical properties of actin and tropomyosin, the untwisting force of actin filaments and the twisting force of tropomyosin filaments act in opposite directions. As a result, the bending stiffness of actin filaments decreases sharply, while that of tropomyosin increases. These forces may promote the sliding of tropomyosin along the actin surface during tropomyosin twisting. During this sliding, the untwisted actin filaments transform into the “M-state”, which facilitates the strong binding of myosin heads. It is interesting to note that greater changes in actin bending stiffness are observed when tropomyosin stiffness is relatively low. It is possible that tropomyosin may modulate the amplitude of actin filament movement relative to thick filaments. If this is the case, the presence of γ-tropomyosin, which maintains a higher intrinsic stiffness, could limit actin filament sliding compared to α- or β-tropomyosins (Table 2). This might relate to the functional role of γ-tropomyosin in slow-twitch muscle fibers, which generate lower contractile force than fast-twitch fibers [105].

When myosin is strongly bound to actin, untwisting forces applied to the actin filament may induce a rotation of the actin filament and the myosin head domain [103]. Conversely, the twisting force exerted on the tropomyosin filament pushes the myosin heads away, potentially causing axial tilting toward the actin filaments (the values Φ_E_ of AEDANS–S1 increase) (Figure 4). In addition, the untwisting force of the actin filaments may be propagated through strongly bound myosin heads to the thick filaments, causing them to rotate in the opposite direction and facilitating the sliding of thin filaments toward the sarcomere center. In this case, tropomyosin may repel the myosin heads from actin, thus contributing to filament sliding. This is typical of thin-filament sliding relative to thick filaments during force production during isometric contraction. It should be noted that the rotation of thick filaments during muscle contraction has been suggested in an earlier study [106]. Our data, obtained using polarization fluorimetry during contraction modeling, also support the possibility of the opposite-direction rotation of thin and thick filaments in contracting muscles [64].

During the transition from the AM•ADP to the AM stage of the ATPase cycle, myosin heads in the absence of nucleotide are capable of interacting with both actin and tropomyosin filaments [101]. Our data show that, compared to the AM•ADP stage, this transition causes an increase in the Φ_E_ angle values for actin filaments containing α-, β-, or γ-tropomyosin by 0.2 ± 0.1, 2.5 ± 0.2, and 0.4 ± 0.1 degrees, respectively (*p* < 0.05). For AF-Tpm, the Φ_E_ values decrease by 2.6 ± 0.2 and 0.7 ± 0.1 degrees for the β- and γ-isoform and increase by 1.3 ± 0.2 degrees for the α-isoform (*p* < 0.05; Figure 2).

The bending stiffness of FITC-actin ε_Actin_ decreases by 7–17%, depending on tropomyosin isoform α, β, or γ, from 4.1 × 10^–26^ to 3.9 × 10^–26^ Nm^2^, from 7.0 × 10^–26^ to 6.3 × 10^–26^ Nm^2^, and from 5.0 × 10^–26^ to 4.2 × 10^–26^ Nm^2^, respectively (*p* < 0.05). In contrast, the bending stiffness ε_Tpm_ for α- and β-isoforms decreases from 13.5 × 10^–26^ to 10.9 × 10^–26^ Nm^2^ (by 19%) and from 13.0 × 10^–26^ to 10.5 × 10^–26^ Nm^2^ (by 19%), and for the γ-isoform increases from 14.3 × 10^–26^ to 16.1 × 10^–26^ Nm^2^ (by 13%), respectively (*p* < 0.05). As a result, the ε_Tpm_/ε_Actin_ ratio decreases for α- and β-tropomyosin from 3.2 to 2.8 (by 14%) and from 1.9 to 1.6 (by 12%), respectively, while, for γ-tropomyosin, it increases from 2.8 to 3.8 (by 35%) (Table 2). Apparently, the release of Pi in the presence of β- and γ-tropomyosins in muscle fibers may be accompanied by a decrease in Pi release capacity. In contrast, for muscle fibers containing α-tropomyosin, where Pi release inhibition is not observed, the ε_Tpm_/ε_Actin_ ratio increases (Table 2). These findings may be attributed to functional differences between muscles fibers with different isoforms of tropomyosin.

Our data also indicate that, during the transition from the AM•ADP to AM stage, a slight deviation of myosin heads from the surface of actin filaments was observed in muscle fibers containing α- or β-tropomyosin (Figure 4). This transition was accompanied by a decrease in the bending stiffness of α-and β-tropomyosins (Table 2). In contrast, during the simulation of Pi release, a pronounced inclination of the myosin heads toward the actin filament axis was revealed (Figure 4). In this case, the bending stiffness of tropomyosin increases rather than decreases, as was also the case during the transformation into the AM stage. Under similar experimental conditions (simulating Pi release, and the transition from AM*•ATP to AM•ADP), previous studies have reported conformational changes in the myosin lever arm (tail), which is thought to be essential for force generation by muscle fibers [107]. However, no notable lever arm movements were detected during the AM•ADP to AM transition, [99]. These observations suggest a correlation between the inclination angle of the myosin head and the movement of its lever arm. Notably, in fibers containing γ-tropomyosin, myosin head deviation was minimal (Figure 4), and a significant increase in tropomyosin stiffness was detected (Table 2). This means that the deviation of the myosin head from the actin filament axis and the inhibition of the lever arm movement of the myosin head depends on changes in the bending stiffness of tropomyosin filaments (Figure 2 and Table 2). From these data, it follows that tropomyosin filaments are involved in the force generation process. Meanwhile, differences in the effects of different types of myosin on the deviation of the myosin head from the actin filament axis and the movement of the lever arm of the myosin head can be attributed to functional differences between muscle fibers with different types of tropomyosin.

During the transition from the AM*•ATP to AM stage, the bending stiffness of actin filaments ε_Actin_ containing α-, β-, and γ-tropomyosins decreases from 7.6 × 10^–26^ Nm^2^ to 3.9 × 10^–26^ Nm^2^ (by 49%), from 7.8 × 10^–26^ Nm^2^ to 6.3 × 10^–26^ Nm^2^ (by 20%), and from 5.5 × 10^–26^ Nm^2^ to 4.2 × 10^–26^ Nm^2^ (by 24%), respectively (*p* < 0.05). The bending stiffness of tropomyosin strands ε_Tpm_ for the α-isoform decreases from 12.4 × 10^–26^ Nm^2^ to 10.9 × 10^–26^ Nm^2^ (by 11%), and, for the β- and γ-isoforms, increases from 8.7 × 10^–26^ Nm^2^ to 10.5 × 10^–26^ Nm^2^ (by 21%) and from 9.9 × 10^–26^ Nm^2^ to 16.1 × 10^–26^ Nm^2^ (by 62%), respectively (*p* < 0.05). Therefore, the ε_Tpm_/ε_Actin_ ratio significantly increases for α-, β-, and γ-tropomyosin from 1.6 to 2.8 (by 71%), from 1.1 to 1.9 (by 68%), and from 1.8 to 3.8 (by 111%), respectively.

When mimicking muscle fiber relaxation (in the presence of MgATP at low Ca^2+^), myosin heads dissociate or weakly bind to actin. Under these conditions, Φ_E_ values for actin filaments containing α-, β-, or γ-tropomyosins decrease by 0.6 ± 0.1, 2.0 ± 0.2, and 0.7 ± 0.2 degrees, respectively (*p* < 0.05; Figure 2), showing increased actin filament twisting. This is accompanied by an increase in the actin bending stiffness values, ε_Actin_, by 15% and 11% for muscle fibers containing α- and β-tropomyosin and a decrease by 16% for muscle fibers containing γ-tropomyosin (Table 2). Concurrently, the Φ_E_ value for tropomyosin strands increase by 2.2 ± 0.2, 2.7 ± 0.2, and 1.0 ± 0.2 degrees for α-, β-, and γ-tropomyosin, suggesting the occurrence of untwisting. The values of ε_Tpm_ decrease by 33% and 23% for α- and γ-tropomyosins and increases by 20% for β-tropomyosin (Figure 2 and Table 2). This may support the hypothesis that actin filament twisting contributes to the rotation of thick filaments and in sliding them away from the center of thick filaments during relaxation.

The release of Pi is a critical step in the muscle contraction cycle [21]. Accordingly, changes in the bending stiffness of actin and tropomyosin filaments, as well as the degree and direction of their twisting and untwisting during ATP hydrolysis, may serve as indicators of ATP activity levels and force generation. Based on our findings, we postulate that force generation occurs during the rapid untwisting of previously overtwisted actin filaments, accompanied by a sharp drop in bending stiffness. Thin filaments push off from the thick filaments and slide toward the center of the sarcomere. Concurrently, tropomyosin strands twist as they translocate along actin, potentially increasing the efficiency of thin-filament sliding relative to the thick filaments.

Tropomyosin filaments are capable of regulating the amplitude of actin filament sliding during force generation. α-, β-, and γ-tropomyosins, which differ in bending stiffnesses, may modulate the force of actin filament untwisting during sliding, thereby allowing for the actomyosin motor to adapt to the functional requirements of different muscle fiber types. For example, the relatively limited untwisting of actin filaments in the presence of γ-tropomyosin, characterized by high bending stiffness, may support the generation of high-tension with low-amplitude sliding. In contrast, α-tropomyosin exhibits to promote larger sliding amplitudes during actin filament untwisting.

This proposal is further supported by studies on tropomyosin mutations under conditions mimicking different stages of the ATPase cycle. These studies have been shown that such mutations can hinder the twisting behavior and bending stiffness of actin and tropomyosin filaments, ultimately affecting myosin ATPase activity [37,84,85,87,108]. These observations highlight the crucial role of the twisting/untwisting of actin and tropomyosin filaments, their relative sliding, and stiffness modulation in the molecular mechanisms underlying muscle contraction.

### 2.5. Impaired Twisting of Actin and Tropomyosin Filaments During the ATP Cycle May Contribute to Muscle Diseases

Even a single-point mutation in tropomyosin significantly alters the polarized fluorescence parameters of FITC-Φctin and AF-Tpm within the F-actin–Tpm–Tn complex, both in the absence and presence of S1 and nucleotides and depending on the Ca^2+^ concentration (Figure 6). These changes primarily reflect the conformational rearrangements of actin and tropomyosin triggered by the mutant tropomyosin during S1 binding in the presence of troponin at either low or high Ca^2+^ levels in the simulations of different stages of the ATPase cycle [37,84,85,87,108]. We interpret a decrease in the orientation angle Φ_E_ combined with an increase in filament bending stiffness (ε) as an indication of filament twisting. Conversely, an increase in Φ_E_ accompanied by reduced stiffness suggests untwisting (see Section 2.2).

According to Figure 6, polarized microfluorimetry data show that the substitutions E150A and E173A in γ-tropomyosin, and R91G and Q147P in β-tropomyosin, cause abnormally high Ca^2+^-sensitivity of thin filaments in muscle fibers. The Φ_E_ values indicate that the filaments adopt an activated conformation, even at low Ca^2+^ levels. Previous studies of myosin ATPase activity have similarly demonstrated that elevated Ca^2+^ sensitivity results from the R90P and E173A mutations in γ-tropomyosin [37], R91G in β-tropomyosin [109], and R167H in α-tropomyosin [110]. Further analysis in this study reveals that all these mutations alter the twisting dynamics of actin and tropomyosin filaments, their bending stiffness (ε_Actin_ and ε_Tpm_), and the ratio of ε_Tpm_/ε_Actin_ (Table 3), which together contribute to a hypercontractile molecular phenotype across multiple stages of the ATPase cycle.

For instance, at low Ca^2+^ levels and in the absence of S1, γ-Tpm–E150A, β-Tpm–R91G and β-Tpm–Q147P exhibit behavior in the B-state that differs markedly from those of wild-type tropomyosin. Unlike the wild-type, which increases actin filament stiffness ε_Actin_ and promotes filament overtwisting in the B-state, these mutants instead reduce actin stiffness and induce untwisting (Figure 6, Table 3). These structural features closely resemble those typically observed in the activated (Ca^2+^-bound) state of the filament: actin untwisting and reduction in bending stiffness (ε_Actin_) (Figure 7, Table 3). This implies that thin filaments containing the mutant tropomyosin variants R90P, R91G, and Q147P are already in a switched-on state, even at low Ca^2+^ levels. This interpretation is consistent with previous data obtained under conditions that simulate various stages of the ATPase cycle [37,68,84,85,86].

During simulated Pi release at low Ca^2+^ levels—corresponding to the transition from the AM*•ATP to the AM•ADP stage—the bending stiffness of actin filaments ε_Actin_ decreased in fibers containing E150A, R90P, R91G, and E173A mutant tropomyosins, while the bending stiffness of tropomyosin filaments ε_Tpm_ increased (Figure 6, Table 3). As a result, the ratio of ε_Tpm_/ε_Actin_ increased by 12%, 54%, 80%, and 35% for these mutants, respectively. By contrast, under similar experimental conditions, this ratio decreased by 13% for γ-tropomyosin and increased only slightly by 8% for β-tropomyosin (Table 3).

This shift likely reflects the untwisting of actin filaments and the enhanced twisting of tropomyosin strands during the AM*•ATP to AM•ADP transition at low Ca^2+^ (Figure 6, Table 3). Notably, this response deviates from the typical low-Ca^2+^ pattern (actin filament twisting and tropomyosin untwisting; see Section 2.4) and instead resembles the behavior observed under high Ca^2+^ conditions—namely, actin untwisting and tropomyosin twisting (Figure 7).

Thus, the results suggest that the E150A, R90P, E173A, and R91G substitutions in tropomyosin severely impair the normal sliding mechanism of tropomyosin filaments relative to actin. At low Ca^2+^ concentrations, mutant tropomyosin slides along actin filaments in a manner that resembles its typical behavior observed under physiological high-Ca^2+^ conditions. As a result, instead of the expected suppression of actomyosin ATPase activity at low Ca^2+^, a pronounced increase in ATPase activity is observed (Figure 8). Considering that during the transition from the AM•ATP to the AM•ADP stage, the bending stiffness of actin filaments (ε_Actin_) markedly decreases while the stiffness of tropomyosin (ε_Tpm_) increases (Table 3), this altered mechanical response may contribute to the enhanced actin-activated ATPase activity of myosin observed under low Ca^2+^ conditions.

A comparable, although less pronounced, effect on actin and tropomyosin filament twisting (Figure 6) and bending stiffness was observed for the Q147P substitution in β-tropomyosin and the R167H substitution in α-tropomyosin (Table 3). These variants are associated with a modest decrease in the Ca^2+^-sensitivity of thin filaments compared to the more disruptive E150A, R90P, E173A, and R91G mutations, as confirmed by ATPase activity measurements (Figure 8) [37,109,110].

In muscle fibers containing R167H mutant tropomyosin, both ε_Actin_ and ε_Tpm_ values decreased during the transition from the AM*•ATP to the AM*•ADP stage at low Ca^2+^. As a result, the ε_Tpm_/ε_Actin_ ratio increased by 4%. In contrast, this ratio remains unchanged in fibers with wild-type α-tropomyosin under the same conditions (Table 3). Furthermore, R167H tropomyosin induces only a slight increase in actin-activated ATPase activity at low Ca^2+^ [110], unlike the E150A and E173A mutants (Figure 8), which are associated with a marked increase in ε_Tpm_ (Table 3). Notably, muscle fibers containing the R90P, E150A, E173A, and R91G mutant tropomyosins exhibited a substantial increase in ε_Tpm_ during this transition, an effect not observed with the R167H variant (Table 3). These findings suggest that an increase in tropomyosin bending stiffness may contribute significantly to the mechanisms of force generation in muscle fibers (see Section 2.3).

According to the data presented in Table 3, during the ATPase cycle at low Ca^2+^ concentrations in the presence of wild-type tropomyosin, the troponin–tropomyosin complex maintains both actin and tropomyosin in the switched-off state (see above). This state is characterized by the overtwisting of actin filaments and a corresponding increase in their bending stiffness. Conversely, thin filament activation involves tropomyosin untwisting, accompanied by a decrease in its bending stiffness (Table 3). Under these conditions, wild-type tropomyosin enables the strong binding of myosin heads to F-actin (Figure 9). However, the results shown in Table 3 and Figure 6 indicate that troponin, when paired with E150A, R90P, E173A, R167H, or Q147P mutant tropomyosins, fails to switch thin filaments off. Instead, actin filaments and tropomyosin strands remain in a constitutively switched-on conformation, as evidenced by actin untwisting and reduced actin stiffness, along with increased tropomyosin twisting and stiffness. These disruptions in the regulatory mechanisms likely underlie both the elevated Ca^2+^ sensitivity and the abnormally high actomyosin ATPase activity observed in the presence of these mutant tropomyosins under low Ca^2+^ concentrations (Figure 6). Consequently, muscle weakness associated with these mutations may stem from impaired twisting/untwisting dynamics and the abnormal mechanical properties (i.e., altered bending stiffness) of actin and tropomyosin throughout the ATPase cycle.

The interaction between troponin and certain mutant forms of tropomyosin, such as E173A, E150A, R90P and R91G, and Q147P (Table 3), impairs the ability of troponin to efficiently activate thin filaments at high Ca^2+^ concentrations. This impairment arises from troponin’s inability to effectively regulate these mutant tropomyosins, as shown by lower values of the ε_Tpm_ parameter and the ε_Tpm_/ε_Actin_ ratio compared to those observed with wild-type tropomyosin. Conversely, in the presence of R167H or K168E mutant tropomyosins, troponin fails to promote actin filament activation, as evidenced by elevated ε_Actin_ values relative to the wild type, while the ε_Tpm_/ε_Actin_ ratio remains reduced (Table 3). These mutant variants markedly suppress actin-activated myosin ATPase activity at high Ca^2+^ (Figure 8).

Tropomyosin mutations also exert a discernible impact on actin–myosin interactions during simulations of intermediate stages of the ATPase cycle. For example, in muscle fibers containing the R90P, Q147P, R167H, and K168E tropomyosin mutants, both actin and tropomyosin filaments exhibit impaired twisting and untwisting dynamics during the transition from the AM*•ATP to the AM•ADP stage at high Ca^2+^. Under these conditions, the characteristic decrease in ε_Actin_ and increase in ε_Tpm_ are also suppressed, leading to a decrease in ε_Tpm_/ε_Actin_ ratio and diminished myosin ATPase activity (Figure 8). A similar reduction in both the ε_Tpm_/ε_Actin_ ratio and ATPase activity is observed for the E150A mutant. However, in this case, the underlying mechanism involves the inhibited twisting of tropomyosin strands combined with the enhanced untwisting of actin filaments, resulting in increased actin flexibility and decreased tropomyosin stiffness.

In contrast, the E173A and R91G mutations moderately enhance ATPase activity and increase the ε_Tpm_/ε_Actin_ ratio during this transition at high Ca^2+^. For E173A, this effect is driven by a marked increase in tropomyosin strand twisting and bending stiffness. For R91G, the mechanism differs: ATPase activity increases due to the enhanced untwisting and decreased rigidity of actin filaments. Notably, these changes are accompanied by inconsistent shifts in actin stiffness for E173A and minimal alterations in tropomyosin stiffness for R91G.

Taken together, the results indicate that several tropomyosin mutations interfere with Pi release under high Ca^2+^ conditions. However, E173A and R91G mutations retain a limited ability to activate the ATPase cycle (Figure 8). A positive correlation was observed between the ε_Tpm_/ε_Actin_ ratio and actin-activated ATPase activity in the presence of mutant tropomyosins at both low and high Ca^2+^ levels (Table 3, Figure 8). These findings suggest that the ATPase activity reduction caused by certain mutations may stem from the inability of troponin and/or myosin heads to properly induce actin filament untwisting or tropomyosin filament twisting during the ATPase cycle (Table 3).

A comparison of all described mutations revealed that the most pronounced decrease in tropomyosin bending stiffness was associated with proline substitutions (Q147P, R90P). Proline is known to introduce significant local distortions (up to 30%) in α-helices, thereby destabilizing their structure [111]. Such distortion interferes with the uniform twisting and untwisting of tropomyosin, and may even reverse its rotational direction (Figure 6). As a result, tropomyosin fails to acquire the necessary helical rigidity required during force generation. We propose that disruption of the rigid tropomyosin “tunnel” impairs the propagation of coordinated torsional oscillations of actin filaments caused by untwisting, thereby preventing their efficient conversion into directional filament sliding. Under normal conditions, actin untwisting promotes sliding, and sliding, in turn, reinforces further untwisting.

In fibers harboring destabilizing mutants such as Q147P and R90P, tropomyosin becomes excessively flexible to effectively coordinate actin’s motion, resulting in a decreased amplitude of actin filament rotation during stage transitions, despite the retention of actin flexibility. This increased tropomyosin flexibility may still facilitate myosin head binding [112], and the absence of a rigid tropomyosin scaffold might even enhance actin oscillations, thereby making myosin binding sites more accessible. This notion is supported by AEDANS–S1 fluorescence data, which demonstrate the increased tilt angles of myosin heads (Figure 4 and Figure 9).

There are also data in the literature indicating that certain amino acid residues, such as alanine, methionine, leucine, lysine, and glutamate, contribute to the stabilization of the α-helical structure of tropomyosin [113]. We investigated the effects of such substitutions (e.g., E173A, K168E) and found that they increased tropomyosin filament stiffness. Interestingly, a greater tilt in the myosin heads relative to the filament axis was observed during the relaxation modeling than in the rigor state, which contrasts with the effect seen with proline substitutions (Figure 9). We attribute this phenomenon to the inability of the overly rigid tropomyosin to properly untwist upon ATP addition and Ca^2+^ reduction, as would typically occur under physiological conditions. As a result, actin filament twisting is impaired, which hinders myosin head detachment and leads to abnormally strong actomyosin binding, even in the presence of ATP.

Mutant tropomyosins, including E150A, R90P, R91G, E173A, R167H, K168E, E117K, and Q147P, exert significant effects on the angular orientation of myosin heads during the ATPase cycle. Specifically, the amplitude of changes in Φ_E_ values for AEDANS–S1 during the transition from weak to strong binding to F-actin (i.e., from the MgATP-bound state to the nucleotide-free rigor state at high Ca^2+^) decreased by 22%, 18%, and 14% for the R90P, K168E, and E117K mutants, respectively. Conversely, this amplitude increased by 24%, 18%, and 20% for the R91G, E173A, and R167H variants, respectively.

These findings suggest that mutant tropomyosins may either inhibit [114] or enhance the efficiency of cross-bridge cycling. For certain mutant forms (such as K168E), inhibition of muscle fiber relaxation was observed, likely due to an increased population of myosin heads remaining in a rigor-like state (Figure 8 and Figure 9). The data presented in Table 3 further support the hypothesis that alterations in myosin head orientation are linked to the disrupted twisting and untwisting of actin and tropomyosin filaments during the ATPase cycle, which may ultimately impair the contractile performance of muscle fibers.

To summarize, the regulation of muscle contraction under normal physiological conditions is tightly controlled by calcium ion concentration. At low Ca^2+^ levels, actin filaments adopt an inactive conformation characterized by increased twisting and elevated bending stiffness (ε_Actin_), while tropomyosin becomes untwisted and more flexible. This structural configuration corresponds to the relaxed state of the muscle. Conversely, at high Ca^2+^ concentrations, the process is reversed: actin filaments untwist, tropomyosin twists and stiffens, and myosin heads gain access to their binding sites on actin, enabling contraction.

In the presence of specific tropomyosin mutations, such as E150A, R90P, R91G, and E173A, this regulatory mechanism is disrupted. At low Ca^2+^ levels, instead of remaining in the relaxed state, actin filaments untwist and tropomyosin strands twist, mimicking the structural pattern typical of the active, high-Ca^2+^ state. This is accompanied by a significant increase in the bending stiffness of tropomyosin, indicating premature activation. As a result, thin filaments remain in a constitutively switched-on state, even under low-Ca^2+^ conditions, leading to abnormally high actomyosin ATPase activity. Such aberrant activation may contribute to muscle weakness, spasms, or contractures observed in various myopathies.

Overall, the data obtained for mutant tropomyosin further confirm their essential regulatory role in guiding actin filament sliding. Disruption of tropomyosin function, whether through helical destabilization or excessive stabilization, interferes with the normal mechanical and structural transitions of actin and myosin. In particular, myosin ATPase activity and force generation are critically dependent on the proper twisting/untwisting dynamics and bending stiffness modulation of both actin and tropomyosin throughout the ATPase cycle. Misregulation of these properties can lead to impaired filament rotation and mechanical output, ultimately contributing to muscle dysfunction. These findings underscore the importance of coordinated torsional and elastic transitions in actin and tropomyosin as prerequisites for effective muscle contraction and relaxation. Further studies into the mechanical role of filament torsion will be essential to fully elucidate these regulatory mechanisms.

## 3. Materials and Methods

### 3.1. Use of Experimental Animals

All experiments were performed on single muscle fibers and isolated proteins from the skeletal muscles of rabbits (*Oryctolagus cuniculus*). A single healthy male animal (1 year old, ~3 kg weight, no prior procedures performed), obtained from a licensed breeding facility, was euthanized in accordance with institutional and national guidelines for the care and use of laboratory animals, as previously described [67]. No in vivo interventions were applied. All procedures, including euthanasia and tissue dissection, were conducted at the Institute of Cytology, Russian Academy of Sciences, and were approved by its ethics committee (identification number F18-00380). Rabbits represent a well-established model for skeletal muscle biochemistry, and are commonly used for actomyosin research due to the high degree of sequence identity and structural similarity between rabbit and human muscle protein isoforms. This study was limited by the use of a single animal, and was conducted entirely in vitro; therefore, human endpoints are not applicable. Potential variability between animals was not assessed. The study protocol was not preregistered in a public database.

### 3.2. Purification and Labeling of Rabbit Skeletal Muscle Proteins

Actin, myosin, and troponin were extracted from the fast skeletal muscles of the rabbit’s back and hindlimbs using standard biochemical techniques. Myosin was purified according to a standard protocol [115]. Myosin subfragment-1 (S1), lacking regulatory light chains, was prepared by treating skeletal muscle myosin with α-chymotrypsin (Sigma-Aldrich, St. Louis, MO, USA) at 25 °C for 10 min. The digestion solution contained 0.12 M NaCl, 2 mM EDTA, 1 mM NaN_3_, 10 mM Tris–HCl (pH 6.8), and 10 mg/mL α-chymotrypsin at a weight ratio of 1:333 (enzyme:myosin) [116]. The reaction was stopped by cooling on ice and adding 1 mM PMSF. The protein was then precipitated with ammonium sulfate (75% saturation), centrifugated, and dialyzed against a solution containing 10 mM KCl, 1 mM MgCl_2_, 0.1 mM NaN_3_, 0.1 mM DTT, and 20 mM Tris–HCl (pH 7.5). S1 was labeled with 1,5-IAEDANS, which specifically binds to the Cys707 residue, at a molar ratio of 1:(0.9–1.1), resulting in the fluorescent conjugate AEDANS–S1 [72]. F-actin of ghost fibers was labeled by incubating individual fibers with 40 μM FITC-phalloidin (Sigma-Aldrich, St. Louis, MO, USA) in a solution containing 10 mM KCl, 3 mM MgCl_2_, 6.7 mM K–Na–phosphate buffer (pH 7.0) for 20 min at room temperature [62,67,68].

Actin for ATPase activity measurements was extracted from an acetone-dried powder, according to a previously described protocol [117]. F-actin was stored in a buffer containing 60 mM KCl, 0.2 mM ATP, 0.2 mM CaCl_2_, 1 mM MgCl_2_, 0.2 mM NaN_3_, and 20 mM Tris–HCl (pH 8.0).

Troponin was extracted from an acetone-dried powder obtained from rabbit skeletal muscle, following established protocols [118,119]. Further purification was carried out using a low-pressure chromatography system (BioLogical LP, BioRad, Hercules, CA, USA).

The degree of fluorescent labeling was monitored using spectrophotometry. The functional activity of S1 AEDANS–S1 was evaluated by measuring the release of inorganic phosphate in a solution containing 30 mM KCl, 2.5 mM MgCl_2_, 0.4 mM CaCl_2_, 2 mM DTT, 0.1 mM NaN_3_, and 12 mM Tris–HCl (pH 6.0), in the presence of 7 μM F-actin and 0.5 μM S1 [120].

Chemicals used were purchased from Sigma-Aldrich (Burlington, MA, USA).

### 3.3. Preparation of Recombinant Tropomyosin

All recombinant wild-type and mutant tropomyosins were overexpressed in *Escherichia coli* BL21(DE3) cells using the bacterial expression system from Novagen (Madison, WI, USA). The *TPM1*, *TPM2*, and *TPM3* genes from human and the *TPM1* gene from rat were cloned into expression plasmids using standard molecular biology techniques [109]. Rat α-tropomyosin was selected due to its high sequence homology (95.8%) with the human γ-tropomyosin isoform. Site-directed mutagenesis was performed using a PCR-based protocol (Stratagene, La Jolla, CA, USA) to introduce the following amino acid substitutions: R91G, E117K, and Q147P in human *TPM2*; R90P, E150A, and E173A in human *TPM3*; and R167H and K168E in rat *TPM1* [110]. To compensate for the absence of N-terminal acetylation, which is critical for actin binding, all tropomyosin constructs were engineered to include an N-terminal Met–Ala–Ser extension. Following transformation, plasmid DNA was produced, isolated, and verified using Sanger sequencing to ensure the quality of cloning and mutagenesis. Protein expression was induced by adding 0.5 mM IPTG (neoFroxx, Einhausen, Germany), and the proteins were purified using ion-exchange chromatography [109]. The purity of the preparations was assessed using SDS–PAGE. Fluorescent labeling of tropomyosin was performed with 5-IAF (Molecular Probes, Eugene, OR, USA), targeting cysteine residues (AF-Tpm) [67]. The following conditions were used: protein concentration of 2 mg/mL, a 10-fold molar excess of dye, and incubation at 37 °C for 24 h.

### 3.4. Ghost Fiber Preparation and Thin-Filament Reconstruction

Single muscle fibers were manually isolated from glycerinated *m. psoas* bundles of rabbit in a cold extraction solution containing 100 mM KCl, 1 mM MgCl_2_, 67 mM K, Na–phosphate buffer (pH 7.0), and 50% glycerol. To obtain ghost fibers, individual muscle fibers then incubated for 70–90 min in a high-salt extraction solution containing 800 mM KCl, 1 mM MgCl_2_, 10 mM ATP, 6.7 mM K, Na–phosphate buffer (pH 7.0).

Thin filaments within the ghost fibers were reconstituted using the sequential addition of troponin and tropomyosin. Decoration with S1 was achieved by incubating the fibers in a solution containing 50 mM KCl, 3 mM MgCl_2_, 1 mM DTT, 6.7 mM K, Na–phosphate buffer (pH 7.0) and the respective proteins at final concentrations of 1.0–2.5 mg/mL. Unbound proteins were removed by washing the fibers with the same solution lacking the added protein.

The final composition of the ghost fibers and the purity of the protein preparations were verified using SDS–PAGE. For analysis, approximately 5–10 single fibers were solubilized in Laemmli sample solution, boiled at 95 °C for 5 min, and subjected to SDS-PAGE using 4% stacking gel and a 12% separating gel. Protein bands were quantified using ImageLab software 6.0.1 (Bio-Rad, Hercules, CA, USA), with calibration curves generated for each protein. The average molar ratios of added proteins to actin in the reconstituted ghost fibers were as follows: tropomyosin (wild-type or mutant), 1:6.9 (±0.8); troponin, 1:7.5 (±1.6); and S1, 1:4.1 (±0.7).

### 3.5. Polarized Fluorescence Measurements

Polarized fluorescence measurements were conducted using a flow-through chamber and a polarized fluorimeter, according to a previously described protocol [48]. Fluorescent probes associated with proteins in muscle fibers were excited using a DRSH-250 mercury lamp (250 W, NTK Azimut Photonics, Moscow, Russia) at specific wavelengths: 407 ± 5 nm for AEDANS-S1, and 489 ± 5 nm for AF-TM and FITC-actin. The excitation light passed through a quartz lens and a double-monochromator, and was then split into two polarized beams using a polarizing prism. One of these beams was reflected by a dichroic mirror and directed onto the fiber through a quartz lens (UV 58/0.80). Fluorescence emission was collected using an objective lens, passed through a barrier filter, and then separated by a Wollaston prism into two components, polarized parallel and perpendicular to the fiber axis. Fluorescence intensities were recorded in the 500–600 nm range using two photomultiplier tubes. Four intensity components were measured, ‖ I ‖, ‖ I ⊥, ⊥ I ⊥, and ⊥ I ‖, where ‖ and ⊥ designate polarization direction parallel and perpendicular to the fiber axis. The first index represents the polarization of the excitation light and the second that of the emitted light. The polarization ratios were calculated using the formulas P‖ = (‖ I ‖ − ‖ I ⊥)/(‖ I ‖ + ‖ I ⊥) and P⊥ = (⊥ I ⊥ − ⊥ I ‖)/(⊥ I ⊥ + ⊥ I ‖). Fluorescence intensity data are available from the corresponding author upon request. Data were analyzed using the helix-plus isotropic model, which allowed for the calculation of the orientation angles of the absorption and emission dipoles at the angles Φ_A_ and Φ_E_, the number of disordered probes N, and the bending stiffness ε of actin and tropomyosin filaments [48,120]. Measurements were performed in a buffer containing 1 mM DTT, 10 mM KCl, 3 mM MgCl_2_, 6.7 mM K, Na–phosphate buffer (pH 7.0) with or without nucleotides (3 mM MgADP, 5 mM MgATP, Sigma Aldrich, St. Louis, MO, USA) to simulate different stages of the S1 ATPase cycle. In experiments involving troponin, the solutions also contained either 0.1 mM CaCl_2_ or 2–4 mM EGTA [67]. Data were collected from 10 to 15 individual fibers (100–150 measurements across all conditions). Statistical analysis was performed using Student’s *t*-test (SigmaPlot software, version 12.5; *p* < 0.05,).

### 3.6. ATPase Activity Measurement

Actin-activated S1 ATPase activity was measured at 25 °C in a solution containing 5.5 mM MgCl_2_, 1 mM DTT, and 5 mM Tris–HCl (pH 7.0). Each reaction mixture included 7 μM F-actin, 1.25 μM troponin, 1.25 μM tropomyosin, and 0.5 μM S1. The reaction was initiated by the addition of MgATP to a final concentration of 3 mM, and was terminated after 10 min by adding trichloroacetic acid to a final concentration of 5%. The amount of inorganic phosphate released was determined colorimetrically, as described previously [119]. Basal ATPase activity of S1 (in the absence of actin) was also measured as a control. To assess Ca^2+^ dependence, the actin–S1 ATPase activity in the presence of troponin was determined using a Ca^2+^/EGTA buffering system, with free Ca^2+^ concentrations corresponding to pCa values from 8 to 4. Each condition was measured in triplicate. The pCa at half-maximal activity (pCa_50_) was calculated using GraphPad Prism 5 (GraphPad Software, San Diego, CA, USA). All datasets were normalized to their respective maximal ATPase activity.

## 4. Conclusions

The troponin–tropomyosin complex regulates actin—myosin interactions in a Ca^2+^-dependent manner by modulating the twisting and bending stiffness of actin and tropomyosin filaments. This process involves the sliding of tropomyosin strands along actin filaments in response to structural changes in the troponin complex.The bending stiffness of actin and tropomyosin filaments is determined by their degree and direction of helical twisting. The untwisting of actin or tropomyosin filaments results in reduced stiffness, whereas increased twisting enhances stiffness.At low Ca^2+^ concentrations (blocked state), Ca^2+^ dissociates from troponin C, triggering a conformational rearrangement within the troponin complex. Troponin I binds to actin, inducing the twisting of actin filaments and increasing their bending stiffness. This is accompanied by structural changes in actin subdomains 1 and 2, rendering the myosin binding site sterically and conformationally unfavorable. Simultaneously, tropomyosin filaments become untwisted and more flexible, allowing for them to oscillate across and obstruct the myosin binding site.At high Ca^2+^ concentrations (closed state), Ca^2+^ binding to troponin C causes troponin I to release actin and interact with tropomyosin. Consequently, actin filaments untwist and become more flexible, whereas tropomyosin filaments twist and stiffen under the influence of troponin I. This coordinated twisting facilitates the azimuthal movement of tropomyosin, exposing the myosin binding sites on actin filaments.Actin and tropomyosin, in coordination with myosin heads, are involved in the molecular mechanism of force generation in muscle fibers.Throughout the ATP cycle, myosin heads modulate the structure and stiffness of thin filaments in a Ca^2+^- and nucleotide-dependent manner, either promoting or suppressing thin filament activation.In the presence of MgATP and Ca^2+^ (AM*•ATP stage), myosin heads promote actin filament overtwisting and stiffening. This simultaneously causes tropomyosin filaments to untwist and become more flexible. In this state, myosin heads tilt away from the actin filament axis and exhibit high mobility.During Pi release (transition from the AM*•ATP to the AM•ADP at high Ca^2+^), myosin binds strongly to actin and weakly to tropomyosin. This causes tropomyosin to twist and stiffen, while actin filaments sharply untwist and become flexible. As myosin rotates toward the actin filament axis, its mobility is reduced. The untwisting torque generated by actin filaments is transmitted through the converter domain and lever arm of myosin, driving thin-filament displacement along thick filaments. This rotational untwisting force causes the rotation of thick filaments, further contributing to filament sliding. Tropomyosin, being twisted and rigid at this stage, assists in force transmission by pushing actin filaments away from thick filaments as they slide toward the thick filament’s center.The transition from the AM•ADP to the rigor AM stage depends on tropomyosin isoforms. With α- and β-tropomyosin, this transition is accompanied by reduced myosin head tilt, decreased actin untwisting, and tropomyosin strand twisting. At the same time, both filaments become more flexible, leading to reduced filament sliding and force production. Conversely, with γ-tropomyosin, the same transition enhances myosin head tilt, increases actin untwisting and tropomyosin twisting, and leads to the increased bending stiffness of tropomyosin and decreased stiffness of actin filaments. This activates filament sliding and greater contractile force.Muscle relaxation involves the detachment or weakening of myosin–actin interactions. This leads to actin filament twisting and tropomyosin filament untwisting. Consequently, actin filaments become more rigid, while tropomyosin filaments become more flexible. The differential in twisting and bending stiffness cause tropomyosins strands to slide over actin, altering the configuration of the myosin binding sites and making it less suitable for myosin attachment. The resulting structural state enables actin filaments to push away from thick filaments, enabling their sliding away from the center of the thick filaments.Muscle contraction is driven by a dynamic interplay of torsional forces and filament rigidity. The twisting and untwisting of actin and tropomyosin filaments, regulated by Ca^2+^ binding and the myosin ATPase cycle, produce the mechanical basis of force generation, as well as force generation due to the active working stroke in the myosin motor. Disruptions in this finely tuned mechanism may lead to contractile dysfunction and muscle pathology.

## Figures and Tables

**Figure 1 ijms-26-06705-f001:**
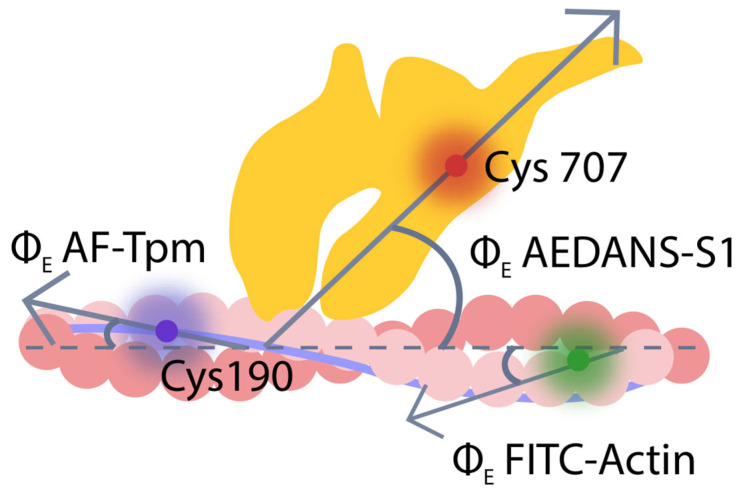
The localization of the fluorescent probes 5-IAF (blue), FITC-phalloidin (green), and 1,5-IAEDANS (red) bound to tropomyosin (Tpm, blue), F-actin (pink), and myosin subfragment-1 (S1, yellow), respectively, in the reconstructed muscle fibers. Gray arrows indicate the direction of polarized light emitted by the fluorescent probe, forming an angle Φ_E_ (emission angle) relative to the fiber axis (dashed line).

**Figure 2 ijms-26-06705-f002:**
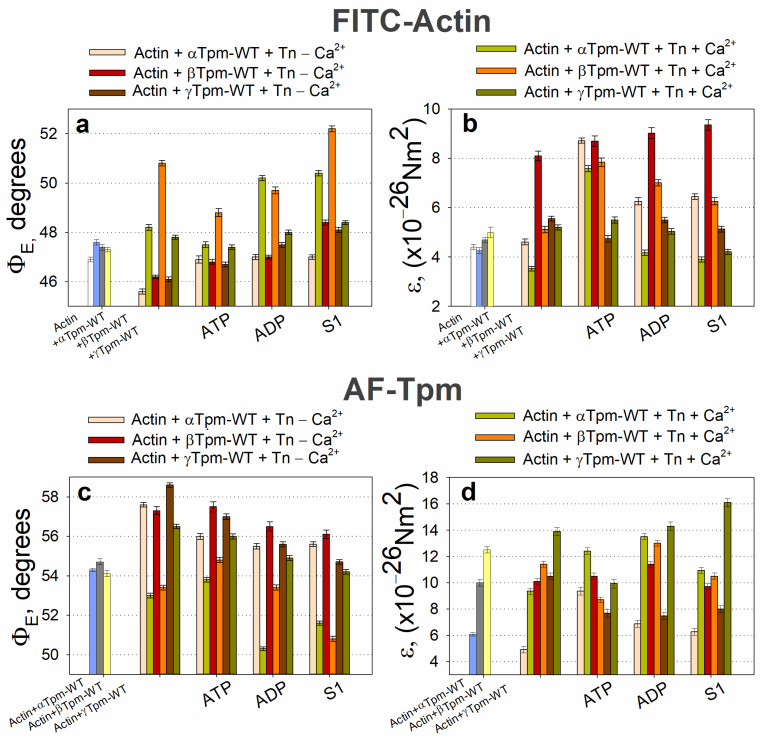
The effect of different tropomyosin isoforms and troponin, as well as Ca^2+^ and nucleotides, on the emission dipole angle (Φ_E_; (**a**,**c**)) and bending stiffness (ε; (**b**,**d**)) of polarized fluorescence from FITC-actin (**a**,**b**) and AF-Tpm (**c**,**d**), respectively, while mimicking different stages of the actomyosin ATPase cycle. Error bars indicate ± SEM. A decrease in the emission dipole angle Φ_E_ is associated with an increase in the twisting of FITC-actin and AF-Tpm. Conversely, a decrease in this angle indicates filament unwinding. An increase or decrease in the ε value indicates an increase or decrease in filament rigidity, respectively. Evidently, low Ca^2+^ increases the twisting and rigidity of actin and has the opposite effect on tropomyosin. Conversely, at high Ca^2+^, actin filament untwists and becomes flexible, while tropomyosin filaments twist and become rigid. In the AM*•ATP stage (in the presence of ATP and high Ca^2+^), the myosin head makes the actin filaments overtwisted, thereby increasing their rigidity. In this case, however, tropomyosin unwinds and becomes flexible. The transition from the AM*•ATP stage to the AM•ADP stage is accompanied by a sharp unwinding of the actin filaments and a decrease in their rigidity. In this case, tropomyosin twists, causing an increase in filament rigidity. Different tropomyosin isoforms can modify this effect in different ways.

**Figure 3 ijms-26-06705-f003:**
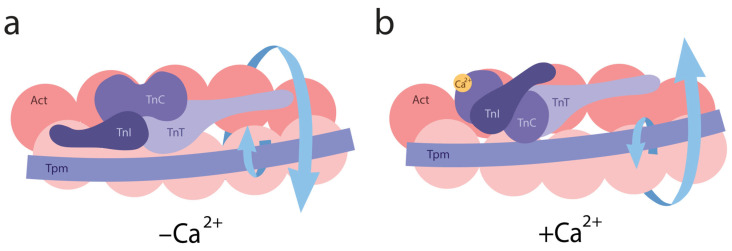
A schematic representation of conformational changes in the helical structures of actin and tropomyosin induced by troponin and calcium ions. (**a**) Arrows indicate the direction of actin twisting and tropomyosin untwisting at low Ca^2+^ levels. (**b**) Upon Ca^2+^ binding to troponin, arrows indicate actin untwisting and tropomyosin twisting. All structural transitions shown in the figure are schematic and not drawn to scale.

**Figure 4 ijms-26-06705-f004:**
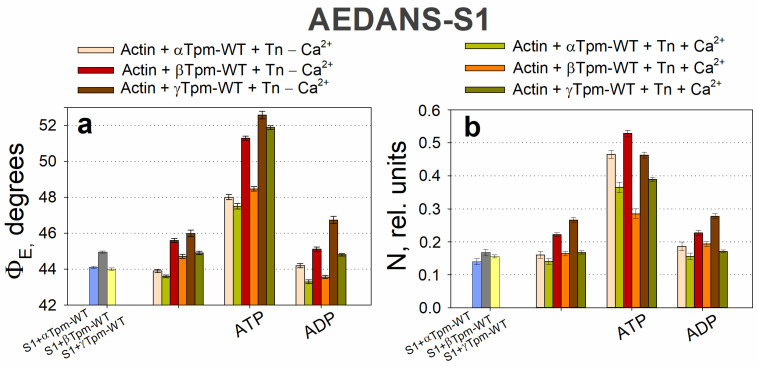
The effect of troponin and Ca^2+^ on the orientation angle of the emission dipoles Φ_E_ (**a**) and the number of disordered probes N (**b**) of the polarized fluorescence of AEDANS–S1 in the presence of α-, β-, or γ-Tpm revealed in ghost fibers under conditions simulating the sequential steps of the actomyosin ATPase cycle. A decrease in the emission dipole angle Φ_E_ is associated with an incline of S1 in the fiber axis. Conversely, an increase in this value indicates S1 deviation from the axis. The N value demonstrates the number of disordered fluorophores, indicating the number of unbound S1. At the AM*•ATP stage, the myosin head weakly bound to actin and deviated from its axis. When transitioning to the AM•ADP and AM stages, the myosin head binds more strongly and tilts toward the actin axis. At the same time, the number of unorganized fluorophores decreases, since more myosin molecules are strongly bound to actin. Data represent the means for 5–7 individual fibers per each experimental condition. Error bars indicate ± SEM.

**Figure 5 ijms-26-06705-f005:**
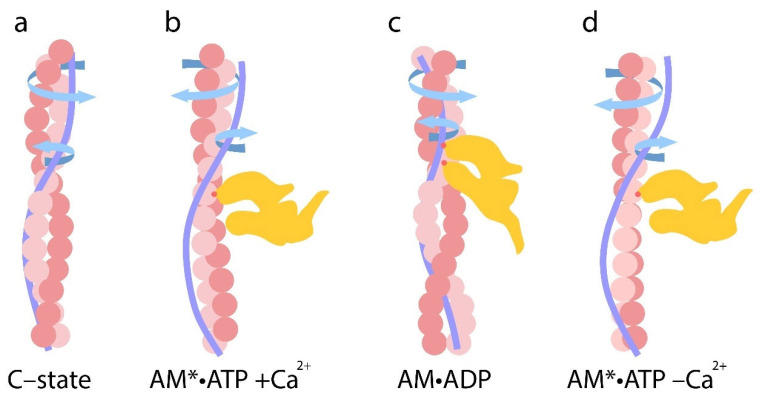
(**a**–**d**) Sequential changes in filament twisting and untwisting at different stages of the ATPase cycle. (**a**) Upon Ca^2+^ binds to troponin, actin and tropomyosin filaments become twisted and untwisted, and their bending stiffness increases and decreases, respectively. (**b**) In the AM*•ATP stage, actin filaments become overtwisted and their bending stiffness increases, while tropomyosin strands undergo untwisting and their bending stiffness decreases. (**c**) During the transition to the AM•ADP stage, strong binding of the myosin head to actin is accompanied by actin untwisting and tropomyosin twisting. Correspondingly, actin stiffness decreases and tropomyosin stiffness increases. The extent of these changes correlates with the level of myosin ATPase activity. (**d**) In the relaxed state (in the presence of MgATP at low Ca^2+^), actin filaments exhibit greater twisting and their bending stiffness increases, while tropomyosin becomes untwisted and more flexible. The arrows indicate the direction of twisting or untwisting of the filaments. All structural transitions shown in the figure are schematic and not drawn to scale.

**Figure 6 ijms-26-06705-f006:**
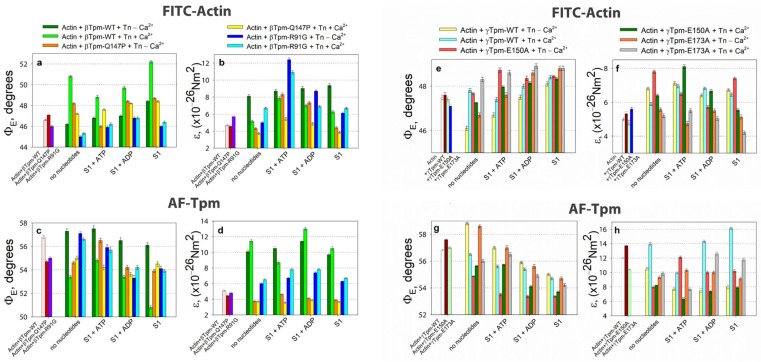
The effect of troponin, Ca^2+^, and nucleotides on the emission dipole angle Φ_E_ (**a**,**c**,**e**,**g**) and bending stiffness ε (**b**,**d**,**f**,**h**) of FITC-actin (**a**,**b**,**e**,**f**) and AF-Tpm (**c**,**d,g,h**), respectively, under conditions mimicking various stages of the actomyosin ATPase cycle in both the absence and presence of mutations in tropomyosin. A decrease in Φ_E_ corresponds to increased filament twisting, while an increase in this angle indicates filament untwisting. An increase in the ε value indicates an increase in filament stiffness, and vice versa. Under low Ca^2+^ in the presence of wild-type (WT) tropomyosin, actin filaments become more twisted and rigid, and tropomyosin strands exhibit the opposite effect. Conversely, at high Ca^2+^, actin filaments untwist and become more flexible, whereas tropomyosin filaments twist and become stiffer. In the AM*•ATP stage (in the presence of ATP at high Ca^2+^), myosin binding causes the overtwisting and increased rigidity of actin filaments, while tropomyosin filaments become untwisted and flexible. The transition to the AM•ADP stage is accompanied by sharp actin filament untwisting and reduced stiffness, along with the simultaneous twisting and stiffening of tropomyosin filaments. Tropomyosin isoforms and point mutations can modify these effects in different ways. Error bars indicate ± SEM.

**Figure 7 ijms-26-06705-f007:**
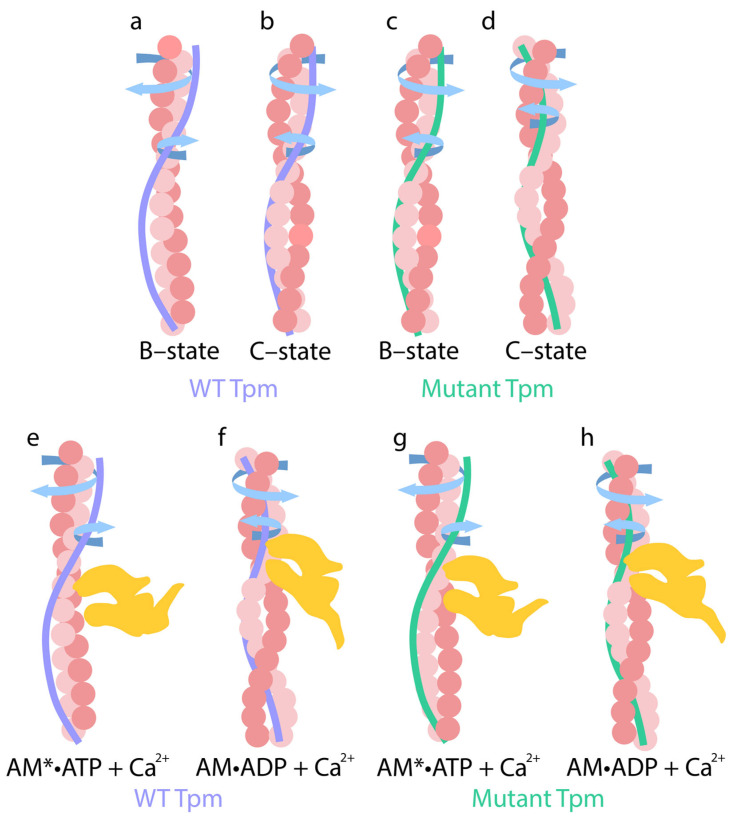
Schematic view of coiling changes in actin and WT (**a**,**b**,**e**,**f**) and mutant (**c**,**d**,**g**,**h**) tropomyosin in the presence of troponin, calcium ions, and nucleotides (ATP, ADP). (**a**,**c**) At low Ca^2+^ concentrations, arrows indicate the direction of actin filament twisting and tropomyosin untwisting. (**b**,**d**) Upon the addition of Ca^2+^, arrows indicate the direction of actin untwisting and tropomyosin twisting. These conformations correspond to the B-state and C-state of the thin filaments, respectively. (**e**) In the AM*•ATP stage, weakly bound myosin heads deviate from F-actin, actin filaments become overtwisted, and their stiffness increases. At the same time, tropomyosin strands become more flexible and untwisted. (**f**) In the AM•ADP stage, myosin heads are inclined to the actin filament and strongly attach to thin filaments, and actin filaments undergo sharp untwisting, accompanied by a significant reduction in their bending stiffness. In contrast, tropomyosin strands twist and become stiffer. (**g**) In the presence of mutant tropomyosin during the AM*ATP stage, myosin heads are more tilted toward the thin filaments. Actin filaments untwist, while tropomyosin filaments twist abnormally. (**h**) At the AM•ADP stage with mutant tropomyosin, myosin heads deviate from the thin filaments. The extent of actin untwisting and tropomyosin twisting is reduced. These changes are accompanied by an increase in actin filament’s stiffness and a decrease in tropomyosin’s stiffness. The extent of these changes correlates with the level of myosin ATPase activity. All structural transitions shown in the figure are schematic and are not drawn to scale.

**Figure 8 ijms-26-06705-f008:**
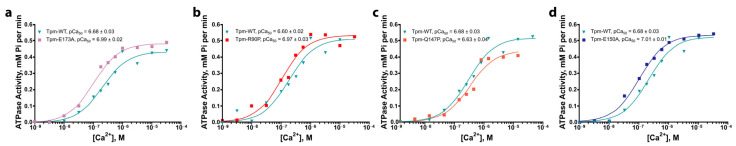
Effects of increasing Ca^2+^ concentration on the actin-activated ATPase activity of S1. Calcium dependence was measured for fully reconstituted thin filaments containing either Tpm–WT or the following mutant variants: Tpm–E173A (**a**), Tpm–R90P (**b**), Tpm–Q147P (**c**), Tpm–E150A (**d**). Representative ATPase activity curves are shown. Average pCa_50_ values from three to four independent experiments are presented; error bars indicate SEM. The shift in the ATPase activity curve to the left indicates the high calcium sensitivity of S1 in the presence of mutant tropomyosins (E173A, R90P, E150), whereas the right shift points to decreased calcium sensitivity in the presence of Tpm-Q147P.

**Figure 9 ijms-26-06705-f009:**
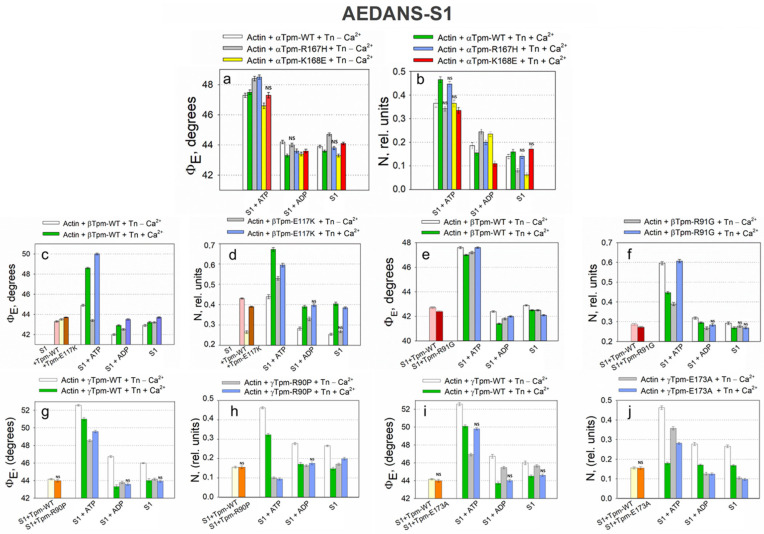
Effects of troponin and Ca^2+^ on the values of Φ_E_ (**a**,**c**,**e**,**g**,**i**) and N (**b**,**d**,**f**,**h**,**j**) measured by the polarized fluorescence of AEDANS-S1 in ghost fibers containing Tpm–R90P (**a**,**b**), Tpm–E173A (**c**,**d**), Tpm–E117K (**e**,**f**), Tpm–R91G (**g**,**h**), and Tpm–R167H and Tpm–K168E (**i**,**j**) under conditions simulating the sequential steps of the actomyosin ATPase cycle. A decrease in the emission dipole angle Φ_E_ is associated with an incline in S1 in the fiber axis. Conversely, an increase in this value indicates S1’s deviation from the axis. All of the mutant tropomyosins, except Tpm-R167H, decrease Φ_E_ on the S1+ATP low-Ca^2+^ state, indicating myosin head inclination and strong binding under relaxation conditions. Normally, myosin heads tilt to the actin axis at the S1+ATP state upon the addition of calcium, as well as upon the further hydrolysis of ATP and the transition to the AM•ADP and AM stages. The presence of mutant tropomyosins caused the opposite effect: the addition of calcium at the S1+ATP state increased rather than decreased Φ_E_, indicating head tilt, and the amplitude of the transition from the AM*ATP to the AM•ADP stage in the presence of calcium ions was lower, indicating reduced force generation. The N value demonstrates the number of disordered fluorophores, indicating the number of unbound S1. Its decrease in S1+ATP in the absence of calcium additionally indicates the increased binding of myosin heads, and an increase in the N parameter in the presence of mutant tropomyosins in the S1+ATP, S1+ADP, and AM stage; high calcium indicates a weakening of myosin head binding during force generation. The index “NS” denotes statistically non-significant differences between wild-type and mutant tropomyosin values of Φ_E_ and N; overwise, differences are considered significant. Error bars indicate ± SEM.

**Table 1 ijms-26-06705-t001:** The effect of troponin and Ca^2+^ binding on the bending stiffness (ε) of wild-type α-, β-, and γ-tropomyosin strands (ε_Tpm_) and actin filaments (ε_Actin_) in reconstructed ghost fibers. Bold values indicate the ratio of tropomyosin to actin stiffness (ε_Tpm_/ε_Actin_).

Tropomyosin Isoform	Presence of High Ca^2+^	Actin–Tpm–Tn		
		ε_Tpm_	ε_Actin_	ε_Tpm_/ ε_Actin_
α	−	4.9	4.6	1.1
α	+	9.4	3.5	2.7
β	−	10.1	8.1	1.3
β	+	11.4	5.1	2.2
**γ**	−	10.5	5.6	1.9
**γ**	+	13.9	5.2	2.7

**Table 2 ijms-26-06705-t002:** Effects of troponin, Ca^2+^, S1, and nucleotides on the twisting of actin filaments and tropomyosin strands (Tpm), and their bending stiffness measured in the ghost fibers. Bold values reflect the ratio of tropomyosin and actin stiffness, ε_Tpm_/ε_Actin_.

Isoform of Tpm	Ca^2+^	Troponin			S1, ATP			S1, ADP			S1		
		ε_Tpm_	ε_Actin_	ε_Tpm_/ε_Actin_	ε_Tpm_	ε_Actin_	ε_Tpm_/ε_Actin_	ε_Tpm_	ε_Actin_	ε_Tpm_/ε_Actin_	ε_Tpm_	ε_Actin_	ε_Tpm_/ε_Actin_
α (Tpm1.1)	−	4.9	4.6	1.1	9.4	8.7	1.1	6.9	6.3	1.1	6.3	6.5	1.0
	+	9.4	3.5	2.7	12.4	7.6	1.6	13.5	4.2	3.2	10.9	3.9	2.8
β (Tpm2.1)	−	10.1	8.1	1.3	10.5	8.7	1.2	11.4	9.0	1.3	9.7	9.4	1.0
	+	11.4	5.1	2.2	8.7	7.8	1.1	13.0	7.0	1.9	10.5	6.3	1.7
γ (Tpm3.12)	−	10.5	5.6	1.9	7.7	4.7	1.6	7.5	5.5	1.4	8.0	5.1	1.6
	+	13.9	5.2	2.7	9.9	5.5	1.8	14.3	5.0	2.8	16.1	4.2	3.8

**Table 3 ijms-26-06705-t003:** The effect of troponin, Ca^2+^, S1, and nucleotides on the bending stiffness of actin filaments (ε_Actin_) and tropomyosin strands (ε_Tpm_) in ghost fibers containing either wild-type (WT) or mutant tropomyosin variants (E150A, R90P, R91G, E173A, R167H, K168E, and Q147P). Bold values reflect the ratio of tropomyosin to actin stiffness, ε_Tpm_/ε_Actin_.

Tpm	Ca^2+^	Troponin			S1, ATP			S1, ADP			S1		
		ε_Tpm_	ε_Actin_	ε_Tpm/_ε_Actin_	ε_Tpm_	ε_Actin_	ε_Tpm/_ε_Actin_	ε_Tpm_	ε_Actin_	ε_Tpm/_ε_Actin_	ε_Tpm_	ε_Actin_	ε_Tpm/_ ε_Actin_
WT 3.12	−	10.5	5.6	1.9	7.7	4.7	1.6	7.5	5.5	1.4	8.0	5.1	1.6
R90P	−	9.0	4.7	1.9	5.4	5.1	1.1	7.8	4.5	1.7	8.1	4.7	1.7
E150A	−	8.0	7.8	1.0	12.1	7.4	1.6	10	5.7	1.8	10.2	7.4	1.4
E173A	−	9.4	6.0	1.6	10.0	5.9	1.7	10.3	4.5	2.3	9.1	5.4	1.7
WT 2.1	−	10.1	8.1	1.3	10.5	8.7	1.2	11.4	9.0	1.3	9.7	9.4	1.0
R91G	−	6.0	5.0	1.2	6.7	12.4	0.5	7.4	8.7	0.9	6.3	6.1	1.0
Q147P	−	3.8	4.3	0.9	4.7	8.3	0.6	4.1	7.3	0.6	3.9	4.4	0.9
WT	−	4.9	4.6	1.1	9.4	8.7	1.1	6.8	6.3	1.1	6.3	6.5	1.0
R167H	−	11.4	4.9	2.3	17.2	6.3	2.8	16.4	5.6	2.9	17.2	5.4	3.2
K168E	−	11.9	4.6	2.6	10.9	6.1	1.8	20.2	6.5	3.1	17.2	5.9	2.9
WT 3.12	+	13.9	5.2	2.7	9.9	5.5	1.8	14.3	5.0	2.8	16.1	4.2	3.8
R90P	+	8.4	5.1	1.6	3.7	5.6	0.7	4.9	5.3	0.9	5.9	5.6	1.1
E150A	+	8.2	6.4	1.3	6.3	8.1	0.8	7.4	6.7	1.1	7.9	5.5	1.4
E173A	+	9.8	5.0	2.3	7.7	4.8	1.6	12.6	4.7	2.7	11.7	4.7	2.5
WT 2.1	+	11.4	5.1	2.2	8.7	7.8	1.1	13.0	7.0	1.9	10.5	6.3	1.7
R91G	+	6.5	6.7	1.0	7.8	10.9	0.7	7.8	6.9	1.3	6.7	6.7	1.1
Q147P	+	3.7	3.7	1.0	3.6	5.4	0.7	3.9	4.9	0.8	3.7	3.9	1.0
WT	+	9.4	3.5	2.7	12.4	7.6	1.6	13.5	4.2	3.2	10.9	3.9	2.8
R167H	+	12.4	3.5	3.6	20.2	5.7	3.5	15.6	3.8	4.1	17.2	3.9	4.4
K168E	+	10.1	3.5	2.9	16.4	5.6	2.9	12.4	3.7	3.3	13.0	3.6	3.6

## Data Availability

Dataset available on request from the authors.

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
