# Peer review of "The Twisting and Untwisting of Actin and Tropomyosin Filaments Are Involved in the Molecular Mechanisms of Muscle Contraction, and Their Disruption Can Result in Muscle Disorders"

_ijms, 2025, doi:10.3390/ijms26146705_

Round 1
Reviewer 1 Report
Comments and Suggestions for Authors
General Comments:
The article utilized the polarized fluorescence microscopy of "ghost" muscle fibers labeled with fluorescent probes (containing F-actin, tropomyosin and myosin) to detect the changes in rotational and bending stiffness of actin and tropomyosin with and without calcium ions, as well as the effects of mutations on twist and untwist. And simulate the role of twist and untwist of the actin and tropomyosin filaments in muscle contraction during the ATPase cycle process. The research and investigation of the article is interesting, but the results and discussion sections are overly lengthy and rather obscure to read. It is recommended to simplify the result analysis and the discussion for greater clarity and conciseness.
Specific comments:
- Introducation: What does AM stand for? Please state clearly.
- Results and discussion: The article is too long. It is suggested to condense the results and reduce the length of the results and discussion sections.
- Figures: Please modify the captions of Figure 2 to be concise and clear, and unify the method of describing the color of the columns. What changes are there in terms of the angle and bending stiffness? Is there a significant difference?
- Figures: It is recommended that Figures 7, 9, and 12 be synthesized into a schematic diagram illustrating the impact of calcium ions and mutations on the interactions between myosin and actin.
The language can be refined further and the description can be more accessible and understandable.
Author Response
Comments to authors
Comments 1: The article utilized the polarized fluorescence microscopy of "ghost" muscle fibers labeled with fluorescent probes (containing F-actin, tropomyosin and myosin) to detect the changes in rotational and bending stiffness of actin and tropomyosin with and without calcium ions, as well as the effects of mutations on twist and untwist. And simulate the role of twist and untwist of the actin and tropomyosin filaments in muscle contraction during the ATPase cycle process. The research and investigation of the article is interesting, but the results and discussion sections are overly lengthy and rather obscure to read. It is recommended to simplify the result analysis and the discussion for greater clarity and conciseness.
Response 1: We are grateful to the Reviewer for the work have been done in analyzing our manuscript. We are confident that the comments made will significantly improve the quality and the perceptibility of our article.
Specific comments:
Comments 2: Introducation: What does AM stand for? Please state clearly.
Response 2: AM stands for actomyosin in the absence of nucleotides, where A – actin and M – myosin. The corresponding explanation has been added to the text (line 41-42).
Comments 3: Results and discussion: The article is too long. It is suggested to condense the results and reduce the length of the results and discussion sections.
Response 3: We agree with the Reviewer that the manuscript is long. Firstly, the large size of the article is related to the fact that the work contains the first time obtained data indicating that the twisting and untwisting of actin and tropomyosin filaments is involved in the mechanisms of muscle contraction regulation and force generation mechanisms. To make such a conclusion, we had to obtain and analyze a huge experimental material with a large variability of experimental conditions. We investigated the structural and functional state of actin, myosin, and tropomyosin at several stages of the ATPase cycle and checked the effect of different forms of tropomyosin (we studied α, β, and γ isoforms) and different aminoacids substitutions in tropomyosin on these stages. Secondly, the main experimental method (highly sensitive to conformational changes in proteins, the method of polarized fluorescence microscopy) is nontrivial and, apparently, unknown to most potential readers. That is why, in our opinion, it was necessary to describe in detail and immediately explain the data obtained with each fluorescent probe under different experimental conditions. By placing the interpretation of the data in different subsections in the Results and Discussion sections, we showed that changes in the orientation of the myosin head, twisting and bending stiffness of actin and tropomyosin are critical for muscle force production and regulation of muscle contraction; we also showed how tropomyosin mutations can cause muscle dysfunction. To reduce the length of the manuscript, we presented only two mutations for different tropomyosin isoforms and did not introduce the results demonstrating the possibility of rehabilitation of muscle dysfunction, which also confirm the idea of the participation of actin and tropomyosin twisting and untwisting in the mechanisms of muscle contraction. It is hardly possible to simplify and shorten the manuscript without deleting some important experimental data or without causing difficulties in understanding the obtained results. However, we have attempted to shorten the manuscript by deleting two figures (Figures 4 and 6) with their description and discussion and deleting several unimportant paragraphs. We have also significantly reduced the list of references (from 157 to 122). In addition, we have corrected the English and tried to make the text clearer. The authors hope that the Reviewer will agree with our arguments and the shortenings made in the corrected version of our paper.
Comments 3: Figures: Please modify the captions of Figure 2 to be concise and clear, and unify the method of describing the color of the columns. What changes are there in terms of the angle and bending stiffness? Is there a significant difference?
Response 3: Color legends for each figure are provided in the top panel of the graph.
Data on figure 2 demonstrates that ATP binding in the presence of Ca2+ ions resulted in significant (p<0.05) twisting (decrease in ФЕ) of actin with an increase in its rigidity (increase in ε), while ATP hydrolysis and release of phosphate and ADP resulted in actin untwisting with a decrease in its rigidity. ATP hydrolysis was also accompanied by twisting and an increase in the rigidity of all three isoforms (α, β, γ) of tropomyosin (p<0.05).
The description below Figure 2 has been changed, and statistically significant changes have been underlined, see lines 250-262.
Comments 4: Figures: It is recommended that Figures 7, 9, and 12 be synthesized into a schematic diagram illustrating the impact of calcium ions and mutations on the interactions between myosin and actin.
Response 4: The recommendation to present Figures 7, 9, and 12 as a schematic diagram seems uninformative to us, since the current visual presentation of information allows the reader to perceive the data obtained better. In the new version of the article, Figure 5 (7 in previous) is presented separately, since it completes the first part of the article, describing the processes of twisting and untwisting in the norm. Figures 9 and 12 dedicated to mutation effects are combined and presented under number 7. The caption to Figure 7 has also been edited for understanding (lines 1083-1101).
Comments on the Quality of English Language:
Comments 5: The language can be refined further and the description can be more accessible and understandable.
Response 5: We thank the Reviewer for the comment. We have significantly revised the text of the article to improve the English and to make the descriptions clearer.
Reviewer 2 Report
Comments and Suggestions for Authors
This is interesting paper.
The authors aimed to measure fluorescence polarization from ghost skeletal muscle decorated by myosin, tropomyosin and actin, and to show that twisting angle changes of a few degrees occur by calcium, ATP, and myosin binding. They discuss molecular mechanism of muscle contraction.
All the experiments were carefully done. However, I raised a few comments for general readers.
The methods seem complicated. Please show the diagram for fluorescence polarization measurement in the text.
Angle calculation seems model-dependent. Please comment in the text.
The twisting or rotation angle changes, they found, are only a few or less than 10 degrees and too small compared with those in the figures 3, 7, 9, and 10. It is difficult to construct models of muscle contraction. Please comment in the text.
Many cryo-EM structures were published, and showed that tropomyosin twists upon calcium binding and myosin binding. The authors should compare the results between two methods, although they compared the bending stiffness between them. Please comment in the text.
Minor. References seem too many. Are there all important?
Author Response
Comments to authors
Comments 1: This is interesting paper.
The authors aimed to measure fluorescence polarization from ghost skeletal muscle decorated by myosin, tropomyosin and actin, and to show that twisting angle changes of a few degrees occur by calcium, ATP, and myosin binding. They discuss molecular mechanism of muscle contraction.
All the experiments were carefully done. However, I raised a few comments for general readers.
Response 1: We thank the Reviewer for high appreciation of our study and valuable comments and suggestions of how to improve our paper.
Comments 2: The methods seem complicated. Please show the diagram for fluorescence polarization measurement in the text.
Response 2: Figure 1 shows the scheme of the fluorescent labels used and the emission angles (ФE) we measured. We also devoted Section 2.1 to the description of the method, partially rewrote it to make the explanation more accessible, and also indicated which components of fluorescence we measure (lines 232-235).
Comments 3: Angle calculation seems model-dependent. Please comment in the text.
Response 3: The exact name of the calculation model (helical plus isotropic model) has been corrected in the text (line 234-235).
Comments 4: The twisting or rotation angle changes, they found, are only a few or less than 10 degrees and too small compared with those in the figures 3, 7, 9, and 10. It is difficult to construct models of muscle contraction. Please comment in the text.
Response 4: In the figures schematically demonstrating the observed changes (Figures 3, 5 and 7 in the new version), a corresponding comment has been added: “All structural transitions shown in the figure are schematic and not drawn to scale” (lines 369-370, 782-783, 1100-1101).
Comments 5: Many cryo-EM structures were published, and showed that tropomyosin twists upon calcium binding and myosin binding. The authors should compare the results between two methods, although they compared the bending stiffness between them. Please comment in the text.
Response 5: As far as we know, to date no pre-power stroke structure has been obtained by cryo-electron microscopy, since technically this has not yet been achieved. There are only a few structures available:
- actin-tropomyosin;
- actin-tropomyosin-troponin in the B- and C-states;
- actin-tropomyosin-myosin head in the M-state and in the presence of ADP.
These structures were obtained by different authors using various protein isoforms, protein ratios and protocols, which makes comparison of their parameters impossible.
Regarding the structures obtained in the same study:
1) Pirani et al. (2005) (doi.org/10.1016/j.jmb.2004.12.013) showed that no significant differences in helical twist were detected between the B- and C-states. This may be due to sample preparation, where protein complexes are frozen, as well as the process of helical reconstruction, in which all parameters are averaged, and the helical twist and rise usually represent a range of values.
2) Doran et al. (2023) (doi: 10.1085/jgp.202213267) note that for both structures (AM and AM-ADP) the twist search ranged from −166.2 degrees to −167.6 degrees, while the rise search ranged from 27.2 to 27.9 Å. Both the rigor and nucleotide-free reconstructions converged on a twist of −166.8 and a rise of 27.9 Å − meaning the parameters were averaged and the comparison is impossible.
Cryo-EM as a method does not allow capturing the dynamics of conformational changes in the muscle protein complex, since it involves a collection of the single frozen states. Our system represents muscle fiber, into which muscle protein or small molecules can be added and changes in conformation can be observed immediately.
Comments 6: Minor. References seem too many. Are there all important?
Response 6: Some references have been removed, their total number reduced from 157 to 122.